# Using statistical models to depict the response of multi-time scales drought to forest cover change across climate zones

Yan Li[1], Bo Huang[2], and Henning W. Rust[1]

[1]Institute of Meteorology, Freie Universität Berlin, 12165, Berlin, Germany
[2]Industrial Ecology Programme, Department of Energy and Process Engineering, Norwegian University of Science and Technology, 7491, Trondheim, Norway

**Correspondence:** Yan Li (yan.li@met.fu-berlin.de)

**Abstract.**

The interaction between forest and climate exhibits regional differences due to a variety of biophysical mechanisms. Observational and modelling studies have investigated the impacts of forested and non-forested areas on a single climate variable, but the influences of forest cover change on a combination of temperature and precipitation (e.g., drought) have not been explored owing to the complex relationship between drought conditions and forests. In this study, we use the historical forest and climate datasets to explore the relationship between forest cover fraction and drought from 1992–2018. A set of linear models and an analysis of variance approach are utilized to investigate the effect of forest cover change, precipitation and temperature on droughts across different time scales and climate zones. Our findings reveal that precipitation is the dominant factor (among the three factors) leading to drought in the equatorial, temperate, and snow regions, while temperature controls drought in the arid region. The impact of forest cover changes on droughts varies under different precipitation and temperature quantiles. Precipitation modulates forest cover's impact on long-term drought in the arid region, while temperature modulates the impact of forest cover changes on both short- and long-term drought in the arid region as well as only on long-term drought in the temperate region. Forest cover can also modulate the impacts of precipitation and temperature on drought. High forest cover leads to a combined effect of precipitation and temperature on long-term drought in arid and snow regions, while precipitation is the only dominant factor in low forest cover conditions. In contrast, low forest cover triggers a strong combined effect of precipitation and temperature on drought in the temperate region. Our findings improve the understanding of the interaction between land cover change and the climate system and further assist decision-makers to modulate land management strategies in different regions in light of climate change mitigation and adaptation.

## 1 Introduction

Forests cover around 4.06 billion hectares, accounting for around 30 % of the global ice-free land surface, and are distributed widely from the tropical to boreal regions (Crowther et al., 2015; Hansen et al., 2013). Global forests have undergone significant changes in the past few decades (Hansen et al., 2013). Most countries have reported a net forest loss due to intensive logging in the tropical region, especially in the 2000s. The tropical forest loss rate increased from -4040 hectares per year in the 1990s to -6535 hectares per year in the 2000s (Kim et al., 2015). And since the early 2000s, more than three-quarters of the Amazon

rainforest has been losing resilience, especially for those in regions with less rainfall and closer to human activity (Boulton et al., 2022). At the mid-latitudes, forest cover fraction has increased owing to accelerated afforestation (Hansen et al., 2013). Large deforestation areas have been detected in the boreal regions due to wildfire occurrence (Hansen et al., 2010). At the national scale, China contributes to the largest afforestation area in the world, as more than 16 sustainability programs have been launched in the country since the 1970s (Bryan et al., 2018). The area of planted forest in China has increased by around 1.7 million hectares per year since the 1990s (Peng et al., 2014). Brazil has been the world leader in tropical deforestation, clearing an average of approximately 1.95 million hectares per year from 1996 to 2005 (Nepstad et al., 2009). Based on five integrated assessment models and Shared Socioeconomic Pathways (SSP) scenarios, global forest areas are likely to decrease up to -600 million hectares in SSP3 (regional rivalry) and increase by up to 1100 million hectares in SSP1 (sustainability) by the end of the 21st century (Popp et al., 2017).

Forests play a vital role in supporting ecosystem services, including local climate regulation via water and heat exchanges with the atmosphere (known as biophysical effects) (Anderson et al., 2011; Bonan, 2008). Changes in forest cover have the potential to alter local climate by affecting surface evapotranspiration, albedo and surface roughness (Alkama and Cescatti, 2016; Mahmood et al., 2014). The net impact on local climate is highly spatially heterogeneous due to the balance among these mechanisms (Perugini et al., 2017; Li et al., 2015; Cherubini et al., 2018). In general, afforestation in the tropics results in regional land surface cooling due to high evapotranspiration, while the effect is warming in the boreal region caused by the typically low surface albedo of forests, especially in the snow-covered winter (Alkama and Cescatti, 2016; Perugini et al., 2017). At mid-latitudes, the effects are more uncertain and have more spatial variability, particularly at a local scale (Mahmood et al., 2014; Perugini et al., 2017; Findell et al., 2017). The uncertainties are mainly caused by competing biogeophysical forcings from albedo and evapotranspiration being similar in magnitude but opposite in sign Bonan (2008). And the regional background conditions (climate and soil moisture), forest types (coniferous vs. deciduous), types of land used for comparison (cropland vs. grassland), or analysis methods (observations vs. climate models) further enlarge such an uncertainty (Ge et al., 2019; Li et al., 2016; Pitman et al., 2011; Tian et al., 2022). For instance, a study using satellite retrieval products shows that the non-radiative (i.e., evapotranspiration) effect dominates in surface cooling in a typical temperate region (i.e. the Loess Plateau in China) (Ge et al., 2019). Nevertheless, for the same region, a coupled land-atmosphere model finds a net surface warming caused by radiative effects (i.e., changes in surface albedo and radiation fluxes) (Tian et al., 2022).

Forest change also can influence the hydrologic cycle by altering evapotranspiration, streamflow, precipitation, and soil moisture (Bonan, 2008; Hoek van Dijke et al., 2022). Deforestation in the tropics can result in a strong decrease in precipitation (Smith et al., 2023), with up to 30 % of annual total precipitation (Snyder et al., 2004; Perugini et al., 2017). This is because more than half of forest evapotranspiration can be recycled and produce rainfall in the tropical region (Salati and Nobre, 1991; Silva Dias et al., 2009). And another reason is that deforestation induces an asymmetrical temperature response in the northern high latitudes and Southern Hemisphere Tropics, which leads to a reduction in the thermal contrast between the hemispheres, resulting in a weakened meridional pressure gradient. Consequently, the convergence of cross-equatorial flows and the precipitation is diminished (Liang et al., 2022). Leite-Filho et al. (2021) revealed that the impact of deforestation on precipitation is intricately linked to the extent of deforestation in the Amazon. Within 28 km grid cells, deforestation can

actually lead to an increase in rainfall, with forest loss of approximately 55-60 % exhibiting this effect. However, once this threshold is surpassed, the decline in rainfall becomes steep. Notably, these thresholds vary when considering larger scales (45–50 % at 56 km and 25–30 % at 112 km grid cells). In the case of 224 km grid cells, rainfall steadily decreases with increasing deforestation. In the boreal region, tree removal leads to a slight reduction in precipitation, around 15% of the annual total precipitation (Snyder et al., 2004; Cherubini et al., 2018). This is likely due to inappreciable differences in the evapotranspiration ratio between different vegetation covers in boreal region (Beringer et al., 2005). Forests' change impacts on precipitation in the temperate regions are more complex than in the boreal or tropical region (Bala et al., 2007; Field et al., 2007; Bonan, 2008). It is challenging to detect the signal of forest cover changes on rainfall in the temperate region, owing to the high variability of synoptic scale meteorological systems, which impact local-to-regional circulation and rainfall patterns (Bala et al., 2007; Field et al., 2007; Bonan, 2008). Modeling studies suggest that a decline in vegetation cover can lead to a reduction in annual precipitation in the temperate region, ranging from -73 to -219 mm per year (Perugini et al., 2017). Based on the Coupled Model Intercomparison Project Phase 6 (CMIP6) results, deforestation has wide-ranging effects on precipitation patterns. It not only causes a significant reduction in local and regional precipitation but also impacts areas beyond the deforested regions. Additionally, deforestation contributes to a decrease in the frequency and intensity of heavy precipitation events while also shortening the duration of the rainy season (Luo et al., 2022).

Changes in temperature and precipitation may affect regional wet and dry conditions, such as drought. Drought is a complex climatic condition characterized by below-normal rainfall over a period from months to years (Dai, 2011). Drought is mainly driven by a combined effect of temperature, precipitation, wind speed, and solar radiation (Seneviratne, 2012). Meanwhile, drought is also considered a natural disaster that poses serious threats to ecosystems by changing the forest structure (tree size, plant life form and potential canopy position) and carbon content (the distribution and types of vegetation that store carbon) (Nepstad et al., 2007). Moreover, several studies have reported changes in drought characteristics in the past few decades (Cook et al., 2014; Trenberth et al., 2014; Naumann et al., 2018; Zhao and Dai, 2015). Since the 1950s, there has been a noticeable trend towards increased drought severity in southern Europe (Vicente-Serrano et al., 2014) and West Africa (Dai, 2013), as indicated by the standardized precipitation evapotranspiration index (SPEI) time series and self-calibrated Palmer drought severity index (scPDSI), respectively. In southern Europe, the trend shows a higher frequency and intensity of droughts, while West Africa has experienced severe to extreme droughts. On the other hand, Western North America has seen a relatively smaller increase in drought severity, with the number of months with moderate to extreme drought showing an increase of around 100 between 1958 and 2008, as compared to the period of 1909-1958, with the maximum value being over 225 months (Peterson et al., 2013).

Some studies have indicated that alterations in forest cover are highly likely to have an impact on regional drought conditions. For example, deforestation leads to less water that can be recycled and intensifies the regional dry seasons in the Amazon region (Bagley et al., 2014; Staal et al., 2020). Based on model simulations, the conversion of mid-latitude natural forests to cropland and pastures is accompanied by an increase in the occurrence frequency of hot-dry summers (Findell et al., 2017). To date, no reports have addressed the impact of forest changes on drought conditions in other regions. Additionally, the different time-scale drought responses to forest change have not been explored. Given the significance of forest management decisions for

climate adaptation and mitigation targets, it is essential to comprehend how drought responds to alterations in forest cover. The statistical model serves as a valuable tool for exploring climate impacts resulting from changes in forest cover (Huang et al., 2023). In this study, we aim to employ statistical models to explore the connection between forest cover changes and drought, keeping the following questions in mind:

1. How do changes in forest cover affect droughts at different time scales?

2. What is the role of forest cover change in modulating drought across various climatic regions?

The objective of this study is to give a fundamental view of the relationship between observational forest cover area shift and drought variation to understand how drought responds to forest change across different time scales and climate zones. A brief introduction involving forest change and its climate effect is given in Section 1. Section 2 presents an overview of the data description and source, followed by a discussion on methods used in the paper in Section 3. Section 4 evaluates the effect of forest cover change on drought in different time scales and climate zones. In Section 4.1, we analyzed the effects of forest cover change and meteorological factors on droughts based on the Analysis of Variance mentioned in Section 3.2. In Section 4.2, we focused on how meteorological factors influence the impact of forest cover change on droughts. Finally, in Section 4.3, we examined the effects of meteorological factors on droughts under extreme values of forest cover area, specifically the maximum and minimum values. Section 5 concludes the main findings along with the limitation and the possible extension of the work.

## 2 Data

### 2.1 Climate Classification

The climate classification is based on the digital Köppen-Geiger World map dataset, which was first formulated by Wladimir Köppen and has been updated for several generations (Kottek et al., 2006; Peel et al., 2007; Kriticos et al., 2012; Beck et al., 2018). The philosophy behind the construction of this version is to rely on observed data rather than experience, wherever possible, to minimize the number of subjective decisions. The map dataset is defined on long-term station records of monthly precipitation sums and monthly mean temperature obtained from the Global Historical Climatology Network (GHCN) version 2.0 dataset (Peterson and Vose, 1997). In the creation of the climate classification dataset, 12,396 precipitation stations and 4,844 temperature stations were used, and there are 30 possible climate types. The latest iteration of the Köppen-Geiger World map dataset, presented by Beck et al. (2018), boasts an exceptional resolution of 0.0083° (roughly equivalent to 1 km at the equator), offering a more precise depiction of regions with high heterogeneity. However, our analysis concentrates solely on five primary climate zones, where we find no substantial distinctions among the available Köppen-Geiger World map datasets. We have classified the 30 climate types into five primary climate groups, namely *equatorial* (11030 grids), *arid* (15673 grids), *temperate* (9587 grids), *snow* (20734 grids), and *polar* (35391 grids) regions, as shown in Fig. 1. However, since forest cover in the polar region is insignificant, this study focuses only on the first four regions.

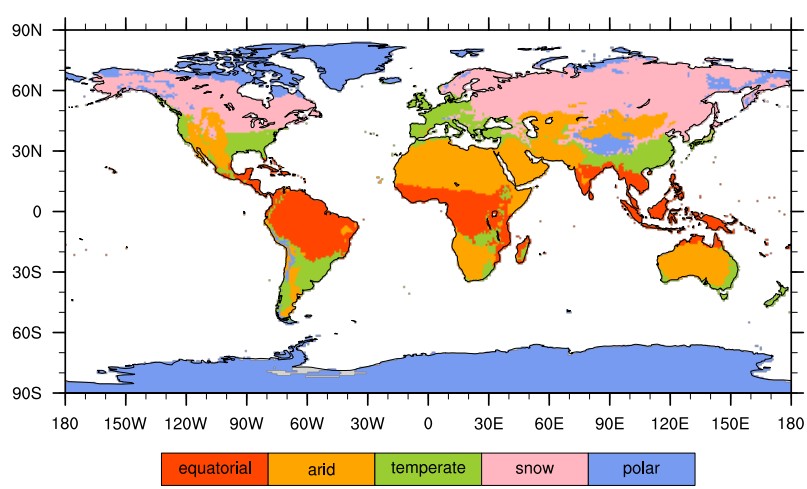

**Figure 1.** Global distribution of main climate classification, according to Köppen- Geiger World map.

## 2.2 Drought Indices

To measure, monitor and analyze the drought, multi indices have been developed (Keyantash and Dracup, 2002). Over the past few decades, there are two widely-used drought indices, the *Standardized Precipitation Evapotranspiration Index* (SPEI) (Vicente-Serrano et al., 2010a) and the *self-calibrating Palmer Drought Severity Index* (scPDSI) (Burke et al., 2006). These two indices describe the effect of temperature and precipitation on droughts. Furthermore, the SPEI describes droughts at different time scales, which is important for our first research question. The SPEI focuses more on atmospheric conditions, while scPDSI considers the situation in the soil. In order to obtain a more comprehensive picture, both indices are used in this study.

### 2.2.1 The Standardized Precipitation Evapotranspiration Index (SPEI)

The SPEI is the extension of the SPI (*Standardized Precipitation Index*), which considers the influence of potential evapotranspiration ($PET$) and uses the difference between precipitation ($Precip$) and $PET$, while SPI maps the precipitation intensity

on a Gaussian variable and uses $Precip$ as the only input (McKee et al., 1993). Similarly to the SPI, SPEI is a time-scale dependent drought index. For this, we specify an integration time scale $\tau$ and a reference month, e.g. SPEI03 for May denotes a drought index obtained for the period from March to May.

$$D_i = Precip_i - PET_i, \tag{1}$$

with $i$ specifying the month. $D_i$ is the water deficit, which can be aggregated for the desired time scale $\tau$. The cumulative $D_i$ is the $D$ series, then the $D$ series is standardized to obtain the SPEI.

The global SPEI dataset used in this study is available as monthly values with a spatial resolution of $0.5° \times 0.5°$. We choose to download various integration time scales $\tau$ of the index: 3 months (SPEI03, short-term), 6 months (SPEI06, mid-term), 12 months (SPEI12, mid-term) and 24 months (SPEI24, long-term). The time period considered is 1992-2018. The calculation of 145 $PET$ is based on the FAO-56 Penman-Monteith method (Allen et al., 1998). And the dataset is based on the Climatic Research Unit (CRU) TS3.24.01 dataset (Vicente-Serrano et al., 2010b, a).

Each grid point is then associated with one of the 5 regions given in Fig. 1. Averaging over all grid points in one region yields the SPEI$\tau$ at monthly resolution for a given region; subsequently averaging over all months of a year yields annual values for each region, e.g., the SPEI03 for the year 2000 is the average over the SPEI03 from January to December of the same year.

**2.2.2 The self-calibrating Palmer Drought Severity Index (scPDSI)**

The Palmer Drought Severity Index (PDSI) is an old drought indicator, developed in 1965 to assess the soil's moisture available by using precipitation and temperature to estimate moisture supply and demand within a two-layer soil model (Wayne, 1965). In 2004, Wells et al. (2004) developed the PDSI into scPDSI, more effectively improving the comparability of the index at different locations. As with the PDSI, the scPDSI is calculated from time series of precipitation and temperature, together 155 with fixed parameters related to the soil/surface characteristics at each location. The fundamental calculation of $PET$ follows Thornthwaite's method (Thornthwaite, 1948). In 2006, Burke et al. (2006) improved the calculation of PDSI, using the Penman-Montheith approach (Maidment et al., 1993) to establish the evapotranspiration, which is applied to the actual vegetation cover, rather than a reference crop (as is done implicitly in the Thornthwaite's method).

The scPDSI dataset (Barichivich et al., 2021; van der Schrier et al., 2013) is available at a $0.5°$ resolution at monthly 160 resolution for the time period 1992-2018. It is based on the CRU TS 4.05 dataset and the calculation of $PET$ is based on the Penman-Montheith method. Each grid point is associated with one of the 5 regions given in Fig. 1. Averaging the scPDSI over all grid points in a region and subsequently averaging all months of the year yields the annual scPDSI for the region.

According to McKee et al. (1993) (both the SPEI and SPI employ the same classification criteria) and Wells et al. (2004), the subsequent classification of the SPEI and scPDSI provides specific ranges for wetness and dryness categories, as outlined 165 in Tab. A2 and Tab. A3. In the case of the SPEI and scPDSI, the normal states are defined as -0.99 to 0.99 and -0.49 to 0.49, respectively. However, in our study, regional averaging was conducted for each index, which tends to normalize the indices

toward the average values. Despite this normalization effect, the positive and negative variations still reveal the tendency of the region to experience dry or wet conditions.

## 2.3 Forest Cover

Changes in forest cover fraction are calculated based on the annual European Space Agency (ESA) Climate Change Initiative (CCI) land cover maps from 1992 to 2018 at a spatial resolution of 300 m (Santoro et al., 2017). The maps describe the earth's terrestrial surface in 37 land cover classes based on the United Nations Land Cover Classification System (UNLCCS), and 14 out of the 37 classes are defined as forest. The dataset was produced after the combination of the global daily surface reflectance of five different satellite observation systems, with the ambition to maintain high levels of consistency over time. It has high

accuracy (>70 %) in representing cropland classes, forests, urban areas, bare areas, water bodies and perennial snow and ice (Poulter et al., 2015). This dataset has been widely used in investigating recent land cover change and its climate effect (Huang et al., 2020; Hu et al., 2020). We aggregated the dataset to a $0.5° \times 0.5°$ resolution, and at each grid point, the forest fraction can take values between 0 and 1. In the following, we use forest fraction aggregated to the level of regions, centered and scaled the annual values by their standard deviation to unit variance for the sake of visualizing the interannual change and comparing

the contribution of forest fraction change in linear models (Sect. 4) to other analogously scaled variables.

Fig. 2 shows the time series of annual forest fraction (centered and scaled to unit variance) for the four regions (equatorial, arid, temperate and snow) as well as the drought indices (scPDSI, SPEI03, SPEI06, SPEI12 and SPEI24) for the period from 1992 to 2018.

In the equatorial region, the forest fraction declines monotonously until 2013 with the decay being slower since 2004. Since

then, it is slightly increasing. From the visual comparison, there is no obvious relation to any of the drought indices. In the arid region, forest fraction declines until 2003 and increases again, interrupted by a few years of decline around 2010. The regeneration to (and even above) values from 1992 happens relatively quickly since 2016. Drought indices are all negative since about 1998 and fluctuate around a relatively constant value since 2010. The temperate zone also shows a decline in forest fraction until 2003, followed by slight ups and downs until 2015. In the last years until 2018, the forest fraction has increased

again to the average level. Compared to the plots in other regions, the drought indices in the snow zone undulate around zero (coordinate on the right y-axis), and there is little difference between SPEI and scPDSI. The forest fraction bars show an almost opposite behavior as in other regions: an increase until around 2009 followed by a decrease leveling off at around 2016.

## 2.4 Precipitation and Temperature

Climatic Research Unit (CRU) monthly near-surface temperature and precipitation datasets with a spatial resolution of $0.5° \times$

$0.5$ ° are used in this study. In the CRU datasets, meteorological station observations were interpolated to grids and given as monthly values by an automated method(Harris et al., 2014). The datasets are used to analyze the effect of meteorological factors on droughts.

Again, for the sake of easier comparison, the variables in Fig. 3 have been centered and scaled to unit variance. Shown is the annual mean of monthly precipitation as bars and temperature as black dots for the different climate zones (different

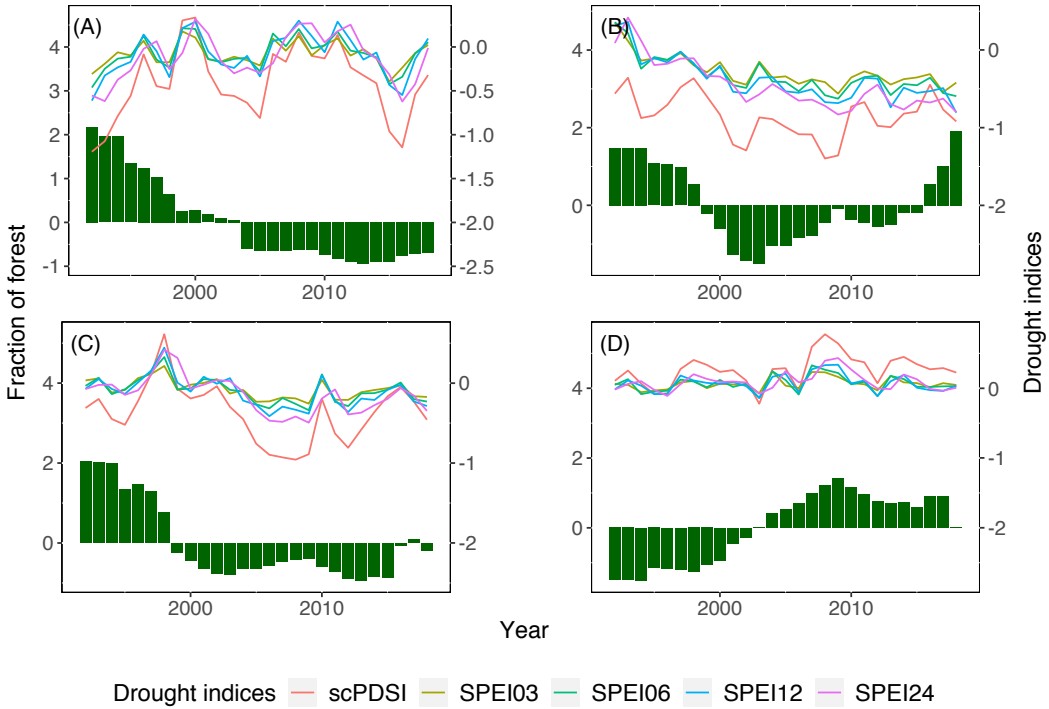

**Figure 2.** Annual change in forest fraction (centered and scaled to unit variance, green bars) and drought indices (colored lines, see legend) from 1992 to 2018 across 4 regions: (A) equatorial; (B) arid; (C) temperate; (D)snow.

columns) in Fig. 3 together with the drought indices considered here (different rows). Note that for the SPEI for 3, 6, 12 and 24 months, temperature and precipitation are analogously aggregated to this length. Hence, for SPEI06 in June, we average monthly precipitation and temperature from January to June and for SPEI06 for July, from February to July and so on. Fig. 3 shows the annual means of precipitation and temperature aggregated analogously to the aggregation level of the drought index.

In equatorial and temperate regions, the main influencing factor to droughts is precipitation, which is largely consistent with changes in the drought index; while this relationship is not visible for the arid region, where the temperature has a larger influence. In snow regions, the situation is even more complex. This implies that precipitation or temperature may not be the dominant factors in this region. Other factors such as their interaction or other environmental variables may play a more important role in driving the changes in drought conditions in the snow region. Furthermore, Fig. 3 also gives an idea about the difference between scPDSI and SPEIs. In equatorial, temperate and snow regions, the change of scPDSI and SPEI is similar; but in the arid region, the variation is different. This is probably due to the low precipitation in the arid region, causing a large difference in the water content of the atmosphere and soil.

Detailed information regarding all the data utilized in this study can be found in Tab. A1.

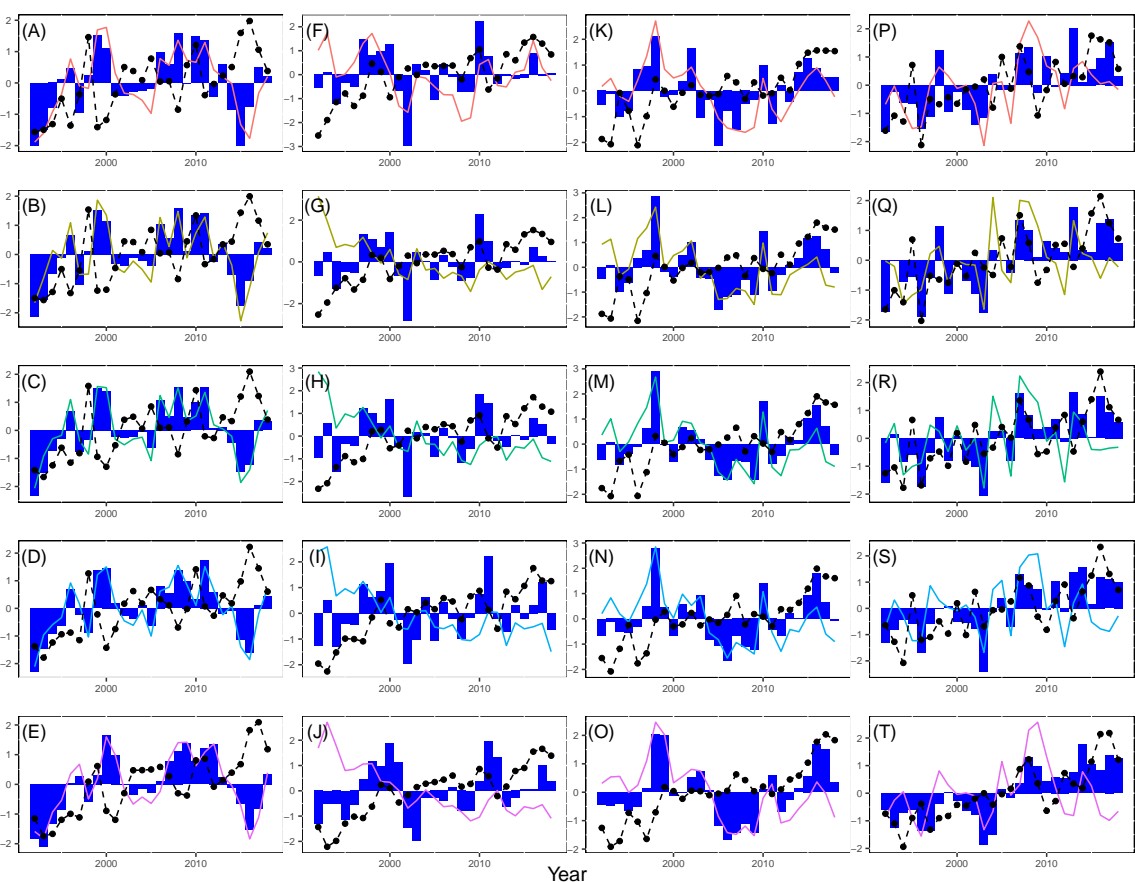

**Figure 3.** Time series of regional averaged annual mean of precipitation (bar), temperature (dashed line) and Drought indices (lines, the legend is the same with Fig. 2; rows from top to bottom: scPDSI and SPEI$\tau$ with $\tau \in \{03, 06, 12, 24\}$) from 1992 to 2018 across different zones (columns from left to right: equatorial, arid, temperate and snow)

## 3 Methods

### 3.1 Linear Models

We use linear models to explore the influence of forest cover, as well as temperature and precipitation on the drought indices in various climate zones. We use linear models because of their great flexibility, versatility and robustness. They are characterized by linearity in parameters to estimate; relations between the predictand and variables in the predictor can still be formulated in a non-linear way. Furthermore, they easily allow to describe joint effects of different variables (temperature and forest cover) on the predictand (interactions), a feature made extensive use of in this study. Linear models allow the assessment of drought

indices for hypothetical situations in a projected climate change scenario. The target variable (drought index) is assumed to be

a realization of a normally distributed random variable $Y$ with constant variance $\sigma^2$ and varying expectation $\mu$

$$Y \sim \mathcal{N}(\mu, \sigma^2). \tag{2}$$

The expectation $\mu$ depends on a set of covariates (or independent variables) $X_1, X_2, X_3, \ldots, X_p$, which we expect to influence the expectation of the target (drought index) in a linear way,

$$\mu = \beta_0 + \beta_1 X_1 + \beta_2 X_2 + \beta_3 X_3 + \ldots \beta_p X_p. \tag{3}$$

The unknown model parameters $\beta_i$ are estimated using maximum-likelihood (Wilks, 2019), realized within the environment for statistical computing R (R Core Team, 2018) using the function `lm()` (Chambers and Hastie, 1992; Wilkinson and Rogers, 1973) from the package `stats`.

For ease of communication, we adopt the notation for linear models introduced by McCullagh and Nelder (1989). For
example, for a model with covariates $X_1, X_2, X_3$ which enter all as direct effects and $X_2 X_3$ as interactions, i.e.

$$\mu = \beta_0 + \beta_1 X_1 + \beta_2 X_2 + \beta_3 X_3 + \beta_4 X_2 X_3, \tag{4}$$

the model notation reads

$$Y \sim X_1 + X_2 * X_3 = X_1 + X_2 + X_3 + X_2 : X_3, \tag{5}$$

with $X_2 * X_3$ being shorthand for $X_2 + X_3 + X_2 : X_3$, i.e. including $X_2$ and $X_3$ as direct effects and as interaction $X_2 : X_3$.
The interpretation of $X_2 : X_3$ is a modulation of the effect of $X_3$ on Y by $X_2$ (or vice versa: modulation of the effect of $X_2$ on Y by $X_3$). Another perspective is to view this as approximating the unknown function using a second-order Taylor expansion, with the resulting unknown parameters estimated from data. By employing this approach to investigate how meteorological conditions and forest cover influence droughts, we aim to generate ideas for potential mechanisms based on data.

Here, we use

$$D_\tau \sim X_{\text{forest}} + X_{\text{precip}} + X_{\text{temp}} + X_{\text{forest}} : X_{\text{precip}} + X_{\text{forest}} : X_{\text{temp}} + X_{\text{precip}} : X_{\text{temp}}, \tag{6}$$

with $D_\tau$ denoting the annual mean of drought index, i.e. scPDSI or SPEI with integration time $\tau$, $X_{\text{forest}}$ the forest cover fraction, for scPDSI, $X_{\text{precip}}$ the annual mean of precipitation and $X_{\text{temp}}$ the annual temperature mean, while for SPEI, $X_{\text{precip}}$ the annual mean of precipitation aggregated similarly to the aggregation level of the drought index and $X_{\text{temp}}$ the annual mean of temperature based on the same method. $X_a : X_b$ denote interactions, e.g. $X_{\text{forest}} : X_{\text{temp}}$ describes the influence of temperature depending on forest cover fraction.
depending on forest cover fraction.

All variables are standardized to zero mean and unit variance before parameter estimation. After building the linear models for each region and drought indices, variance inflation factors (VIF) for each variable in each linear model have been calculated. The values are all less than 5, indicating that the colinearity among these variables can be neglected. The SPEI is a time-scale dependent drought index with integration time $\tau$. Note that the integration time $\tau$ also defines the first available data point for

$D_\tau$ as $\tau - 1$ month after the start of the time series in January 1992. Thus for longer integration times, the SPEI$\tau$ cannot be obtained for the first years of the data set, e.g. for SPEI24, the calculation starts in December of the second year.

Based on Eq. 6, we estimate the impact of forest fraction and meteorological factors on drought. For SPEI03, SPEI06 and SPEI12 of a specific month, we use the previous 3, 6 or 12 months' data; for SPEI24, we do need the data of the previous 2 years.

Fig. 4 compares annual values for the scPDSI and SPEI$\tau$ obtained as described in Sec. 2.2 (points) to the expectation from the linear models (Eq. 6 using forest fraction $X_{\text{forest}}$, temperature $X_{\text{temp}}$ and precipitation $X_{\text{precip}}$ as inputs) (lines); the rows give the scPDSI and the SPEI$\tau$ for different integration time scales $\tau \in \{03, 06, 12, 24\}$ (from top to bottom) and across various climate zones (columns, equatorial, arid, temperate and snow, from left to right). And in Tab. 1, we also give MSE (Mean Squared Error) and Adjusted $R^2$ for all models in Fig. 4.

The Mean Squared Error (MSE) quantifies the difference between model estimates and observed values for the drought indices, computed as,

$$MSE = \frac{1}{n}\sum_{i=1}^{n}(y_i - \widehat{Y_i})^2 = \frac{1}{n}\sum_{i=1}^{n}(D_{\tau,i} - \widehat{D_{\tau,i}})^2 , \tag{7}$$

with $y_i$ representing the observed values for the drought index ($D_{\tau,i}$, scPDSI and SPEIs), $\widehat{Y_i}$ denote the model estimates ($\widehat{D_{\tau,i}}$), $n$ the number of data points, i.e. the number of years. The coefficient of determination $R^2$ gives the fraction of variability described by the model

$$R^2 = \frac{SS_{\text{reg}}}{SS_{\text{tot}}} = 1 - \frac{SS_{\text{res}}}{SS_{\text{tot}}} \tag{8}$$

with the total sum of squares

$$SS_{\text{tot}} = \sum_{i=1}^{n}(y_i - \overline{y_i})^2 = SS_{\text{reg}} + SS_{\text{res}} , \tag{9}$$

the sum-of-squares of the regression

$$SS_{\text{reg}} = \sum_{i=1}^{n}(\widehat{Y_i} - \overline{y_i})^2 , \tag{10}$$

the sum-of-squares of the residuals

$$SS_{\text{res}} = \sum_{i=1}^{n}(y_i - \widehat{Y_i})^2 , \tag{11}$$

and $\overline{y_i}$ denoting the arithmetic mean of the observations $y_i$. We use the adjusted $R^2$

$$R^2_{\text{adj}} = 1 - \frac{SS_{\text{res}}/df_{\text{res}}}{SS_{\text{tot}}/df_{\text{tot}}} = 1 - \frac{\sum_{i=1}^{n}(y_i - \widehat{Y_i})^2/(n-1-p)}{\sum_{i=1}^{n}(y_i - \overline{y_i})^2/(n-1)} , \tag{12}$$

with $df_{\text{tot}}$ and $df_{\text{res}}$ denoting the degrees of freedom for the total and residual sum-of-squares, respectively, $n$ the number of observations and $p$ the number of covariates (independent variables).

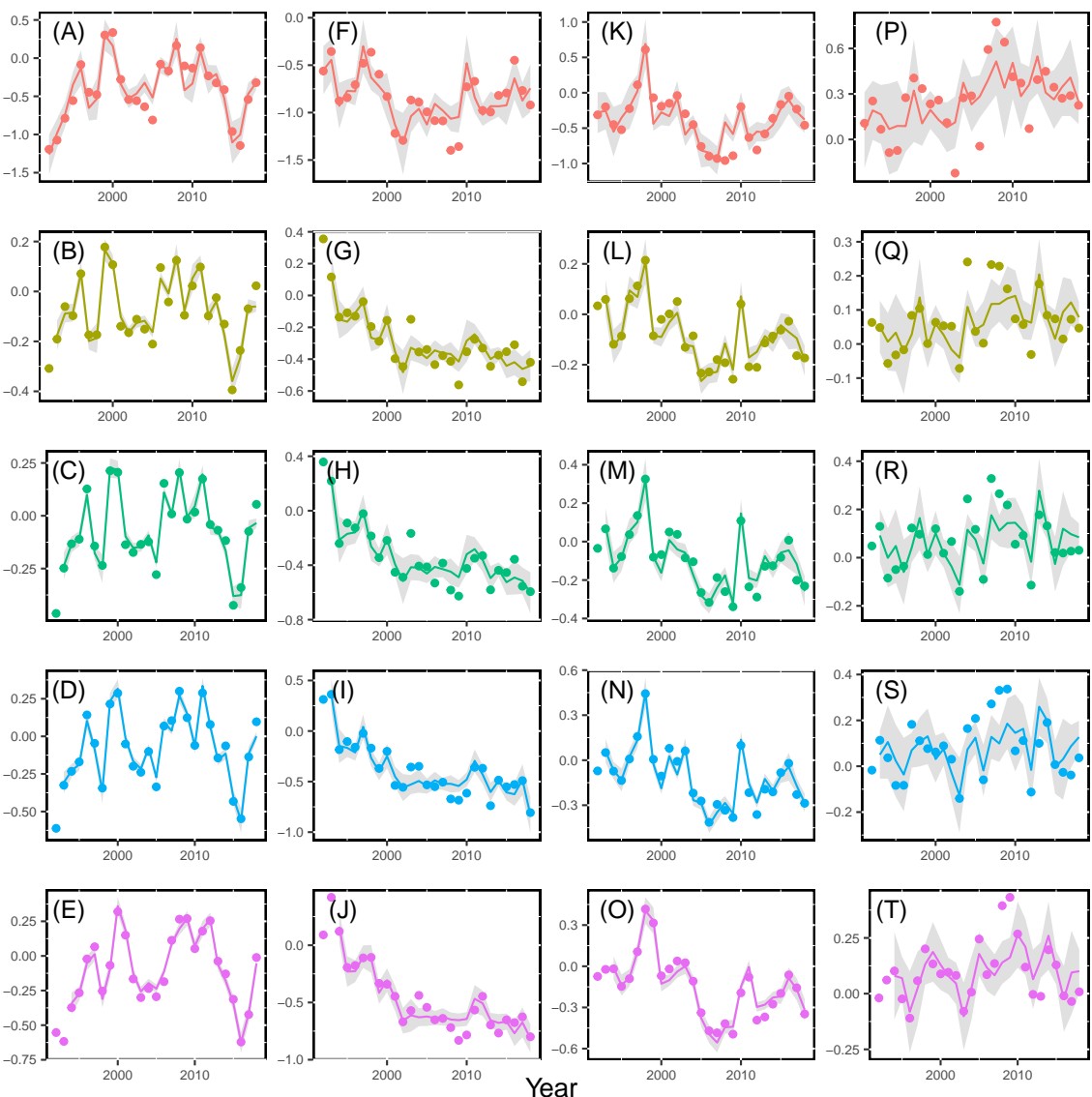

**Figure 4.** Time series of drought indices (colors as same as in Fig. 2); annual point estimates (points) and estimates from the linear model (Eq. 6, lines) across different climate zones (columns from left to right: equatorial, arid, temperate and snow) and for the scPDSI and SPEI$\tau$ with different integration times (rows from top to bottom: scPDSI and SPEI$\tau$ with $\tau \in \{03, 06, 12, 24\}$). The shade signifies the range lies at a 95% level of confidence. Note that SPEI03, SPEI06 and SPEI12 cannot be obtained for the first year and SPEI24 can not be obtained for the first two years.

**Table 1.** MSE (Mean Squared Error) and Adjusted $R^2$ ($R^2_{\mathrm{adj}}$) for all models in Fig. 4.

| | Equatorial | | Arid | | Temperate | | Snow | |
|---|---|---|---|---|---|---|---|---|
| | MSE | $R^2_{\mathrm{adj}}$ | MSE | $R^2_{\mathrm{adj}}$ | MSE | $R^2_{\mathrm{adj}}$ | MSE | $R^2_{\mathrm{adj}}$ |
| scPDSI | 0.12 | 0.84 | 0.31 | 0.59 | 0.23 | 0.70 | 0.57 | 0.23 |
| SPEI03 | 0.05 | 0.93 | 0.17 | 0.77 | 0.10 | 0.86 | 0.49 | 0.32 |
| SPEI06 | 0.03 | 0.95 | 0.17 | 0.76 | 0.09 | 0.88 | 0.45 | 0.39 |
| SPEI12 | 0.03 | 0.97 | 0.11 | 0.85 | 0.05 | 0.94 | 0.53 | 0.27 |
| SPEI24 | 0.02 | 0.97 | 0.09 | 0.87 | 0.04 | 0.94 | 0.53 | 0.27 |

The linear model is able to capture the inter-annual variability of the drought indices to a certain extent with performance varying across climate zones. From visual inspection and comparison of $R^2_{\mathrm{adj}}$, drought indices in the equatorial region can be described best ($0.84 < R^2_{\mathrm{adj}} < 0.97$), while for the snow region ($0.23 < R^2_{\mathrm{adj}} < 0.39$), the model is by far not as performant, cf. Tab. 1. For arid and temperate zones, the models are almost as performant as for the equatorial zone. The MSE in the equatorial region is around 0.1 and smaller, while in the snow region, we find MSE of around 0.5 (Tab. 1). Thus in the equatorial, temperate and arid regions, linear models with two-point interactions of forest cover, temperature and precipitation are well suited to describe the drought indices used here; whereas in the snow region, the factors influencing the drought indices must be more complex than this.

Furthermore, we see from Fig. 4 and Tab. 1 that the SPEI indices with varying time scale $\tau$ are consistently better represented by the model (larger $R^2_{\mathrm{adj}}$) than the scPDSI over all regions, with performance ($R^2_{\mathrm{adj}}$) roughly increasing with $\tau$ (except for the snow region). We hypothesize that this is an effect of the calculation of the scPDSI based on a 2-layer soil box model and thus local soil conditions are relevant; the latter are not represented in our model-building process. Furthermore, we have included residual versus fits plots for all drought indices across different regions in Fig. B1. Upon careful inspection of the residual plots, no evident structures or patterns are observed, indicating that there are no apparent missing terms or heteroscedasticity present in the models. In addition, the significance tests for all linear models have been incorporated into Tab. B1. This table provides the significance stars for p-values associated with each coefficient in the linear regression models.

## 3.2 Analysis of Variance

Analysis of variance (ANOVA Anscombe, 1948) gives a quantitative estimate of the relative strength of these factors, which is used to quantify the effect of the various covariates $X$. We use it to describe the drought indices with, i.e. forest fraction, precipitation and temperature, cf. Eq. 6. We denote the full model for a drought index $D$ as $D_{\mathrm{full}}$ as given in Eq. 6. The model without the information on the forest is denoted as

$$D_{\text{-forest}} \sim X_{\mathrm{precip}} * X_{\mathrm{temp}} , \tag{13}$$

analogously, we denote models without information on temperature or precipitation as $D_{\text{-temp}}$ and $D_{\text{-precip}}$, respectively. With

$$SS_{\text{forest}} = SS_{reg}(D_{\text{full}}) - SS_{reg}(D_{\text{-forest}}), \tag{14}$$

we assess the improvement in model performance in terms of regression sum-of-squares due to including the forest fraction as a covariate, with the $F$-test giving significance to that. Note that the difference in degrees of freedom used in the $F$-test for the full and the reduced model is $\delta p = 3$ as all terms in Eq. 6 involving forest fraction $X_{\text{forest}}$ are taken out. Analogously, we define $SS_{\text{temp}}$ and $SS_{\text{precip}}$ for quantifying the influence of temperature and precipitation.

The fraction of variance contributed to the regression by forest fraction is given as

$$\Delta SS_{\text{forest}} = \frac{SS_{\text{forest}}}{SS_{reg}(D_{\text{full}})}, \tag{15}$$

analogously for precipitation ($\Delta SS_{\text{precip}}$) and temperature ($\Delta SS_{\text{temp}}$).

## 4 Results

For each region, linear models for scPDSI, SPEI03, SPEI06, SPEI12 and SPEI24 are built to describe the relationship between the drought indices ($D_\tau$) and 3 covariates (factors) ($X_{\text{forest}}$, $X_{\text{precip}}$, $X_{\text{temp}}$). The annual values for forest fraction $X_{\text{forest}}$ and the SPEI$\tau$ have a time delay, as discussed in Sec. 3: the model for SPEI03, SPEI06 and SPEI12 uses meteorological data from 1993 to 2018 while the values for forest fraction are from 1992 to 2017; the model for SPEI24 uses meteorological data from 1994 to 2018, while values for forest fraction are from 1992 to 2016; for scPDSI, all values for the covariates are from 1992 to 2018. Here, the results in terms of the proportion of variance added to the models by forest cover change and other meteorological factors across different regions are displayed first. Subsequently, we investigate the interaction of the three factors on droughts in four regions. Key insights emerging from the results are discussed and we also give some possible explanations.

### 4.1 The proportion of variance described by forest cover change

Based on the linear models for all variants of drought indices and across climate zones, we estimate the contribution of the 3 covariates to the regression according to the procedure described in Sec. 3.2. The bars in Fig. 5 show the proportion of variance contributed by forest cover fraction ($X_{\text{forest}}$, green), precipitation ($X_{\text{precip}}$, blue), and temperature ($X_{\text{temp}}$, red) to the regression.

The contribution of precipitation to variability dominates for drought indices in equatorial, temperate, and snow regions. For the arid region, precipitation only dominates scPDSI, while for SPEI-based drought indices, temperature dominates its variability. For the temperate region, precipitation contributes the most considerable fraction of variance compared to the other areas, followed by the snow region.

Across all regions, the forest fraction describes a more significant fraction of the regression variance for scPDSI than for the SPEI-based drought indices. As scPDSI is the only drought index used here that involves soil properties, we hypothesize that forest fraction is linked to these and thus has some potential to describe more about the variability of scPDSI. However, we still should note that the linear models better represent SPEI-based indices (higher $R^2_{\text{adj}}$ in Tab. 1) than scPDSI.

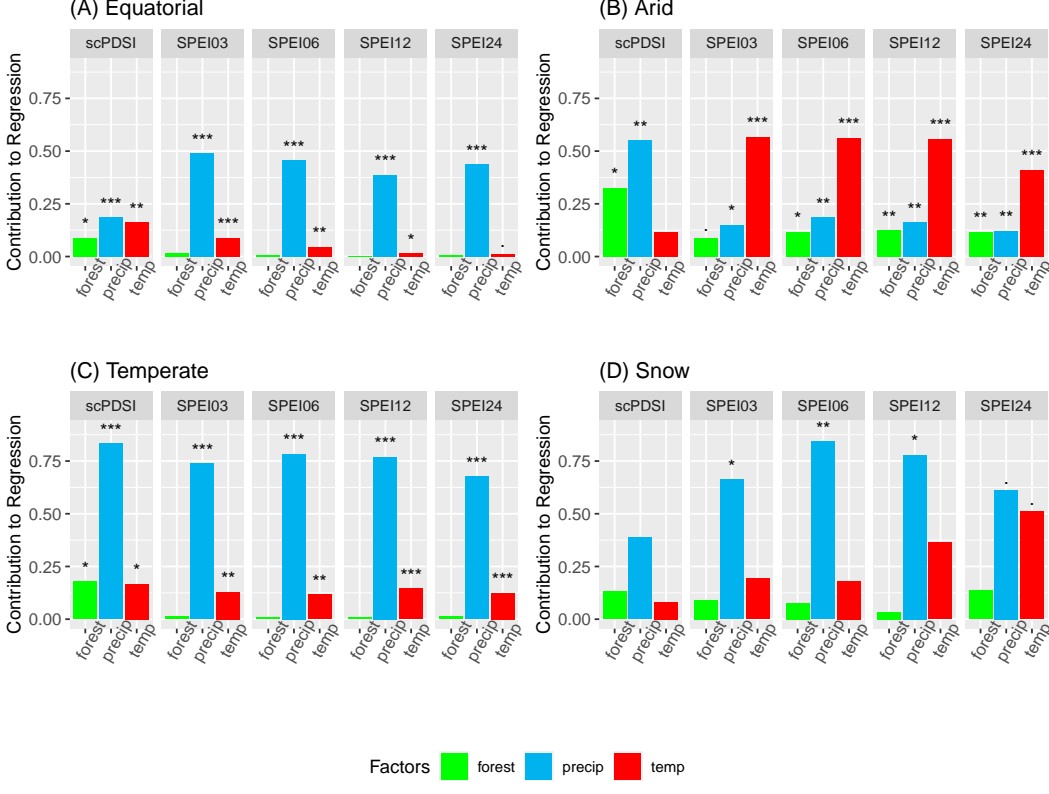

**Figure 5.** Proportion of variance contributed by forest fraction, precipitation, and temperature to the full model that describes drought indices at different time scales across 4 climate regions. (confidence significant level at '***' 0.001, '**' 0.01, '*' 0.05, '.' 0.1.)

For the equatorial and temperate regions, forest fraction does not contribute to describing the variability of the SPEI-based indices (non-significant). In the arid region, however, the proportion of variance contributed by forest cover to SPEI-based indices is around 0.1 and at least significant on the 0.05 level (SPEI03 only on 0.1 level). While for the snow region, the contributing fraction of variance is comparably large but not statistically significant on any reasonable level. Here, the contribution of precipitation is particularly large for SPEI06 and the influence of temperature seems to increase with time scale $\tau$, however, the latter is not significant on the normal levels. Thus for the snow region, precipitation seems to have a larger impact on short-term drought indices while temperature affects more long-term indices. Note that the linear models for the snow region show adjusted $R^2$ values of around 0.3 and hence are a lot less capable of describing the index variability than that in other regions ($R^2_{\text{adj}} \geq 0.7$), cf. Tab. 1. Processes in the snow region seem to be more complex than what can be represented with the approach here.

Proportions of contribution do vary across regions and drought indices: precipitation has a large influence in all regions except the arid region; the temperature has a larger influence on SPEI-based drought indices in the arid region; forest fraction

describes a larger fraction of the regression variance for scPDSI than for the SPEI-based indices. Thus in the arid region, where the ecosystems are fragile and highly vulnerable to climate change, the forest fraction has a stronger impact on changes in drought indices.

## 4.2 Variations in droughts response to forest fraction according to different precipitation and temperature quantiles

Fig. 6 explores the effect of forest cover fraction on SPEI24 and SPEI03 conditioned on precipitation and temperature. Again, this analysis is based on the previously discussed models. The first two rows show the effect of forest cover on SPEIs for different levels of precipitation (dark green lines) with temperature held fixed at its median. For the bottom two rows, precipitation has been held fixed at its median and the forest cover effect for various levels of temperature ( dark yellow lines) is explored. The strength of the colored bands is associated with the quantiles of precipitation/temperature. The dark green/dark yellow part of the band covers the central part of the precipitation/temperature (0.4 to 0.6-quantile). For each successive outer band the quantile level for temperature/precipitation changes by 0.05. Furthermore, similar analyses for SPEI06, SPEI12 and scPDSI can be found in Fig. C3 and Fig. C4 and Fig. C5.

We can see that the influence of forest cover on drought conditions varies depending on precipitation levels and geographical regions. For SPEI03, it appears that forest fraction has a relatively modest impact across various levels of precipitation (lines are close to horizontal). This suggests that precipitation does not significantly modulate the influence of forest cover on the short-term drought index (the first row in Fig. 6). However, when we examine SPEI24, a different pattern emerges. There is a strong influence of precipitation on the forest fraction effect (the second row in Fig. 6), particularly in arid regions (as seen in Fig. 6F). In general, as precipitation increases beyond the median level, the drought index tends to rise with increasing forest cover. This phenomenon can be explained by increased transpiration associated with larger amounts of precipitation, resulting in a reduced vapor pressure deficit (VPD) and, consequently, lower $PET$. This leads to higher SPEI24 values when forests are denser. Furthermore, forests have the capacity to intercept precipitation and diminish ground-level wind speeds. These combined effects contribute to a reduction in $PET$ as forest cover increases. If precipitation is lower, this effect decreases and the slopes in Fig. 6F get smaller. There is not sufficient water to be evapotranspirated, even if the forest fraction increases. For a specific amount of precipitation (about the median) the slope is 0. When the precipitation is less than this amount, we see a negative slope suggesting the interpretation that for restricted water supply, an increase of trees leads to an increase of $PET$ and hence to a decrease of SPEI24. An opposite effect can be observed in the snow region (Fig. 6N). Here, with minimal forest cover, precipitation directly affects the SPEI24 leading to a more humid situation with higher precipitation. However, with forest cover increases, this direct effect vanishes. It's worth noting that for snow regions, the model captures less than 30% of the total variability, indicating the complexity of this relationship.

The modulation of forest cover by temperature (Fig. 6, two bottom rows) is more diverse. In the equatorial region (Fig. 6C and D), the temperature influence is much weaker than precipitation (cf. Fig. 5), hence precipitation dominates the drought index in the equatorial region. However, for the higher temperature, we see a slight decrease in SPEI24 (Fig. 6D) with forest cover, while for the low temperature, SPEI24 increases with forest cover. However, the significance test, as presented in Tab. B1, reveals that the interaction between temperature and forest cover to SPEI24 does not demonstrate statistically significant results.

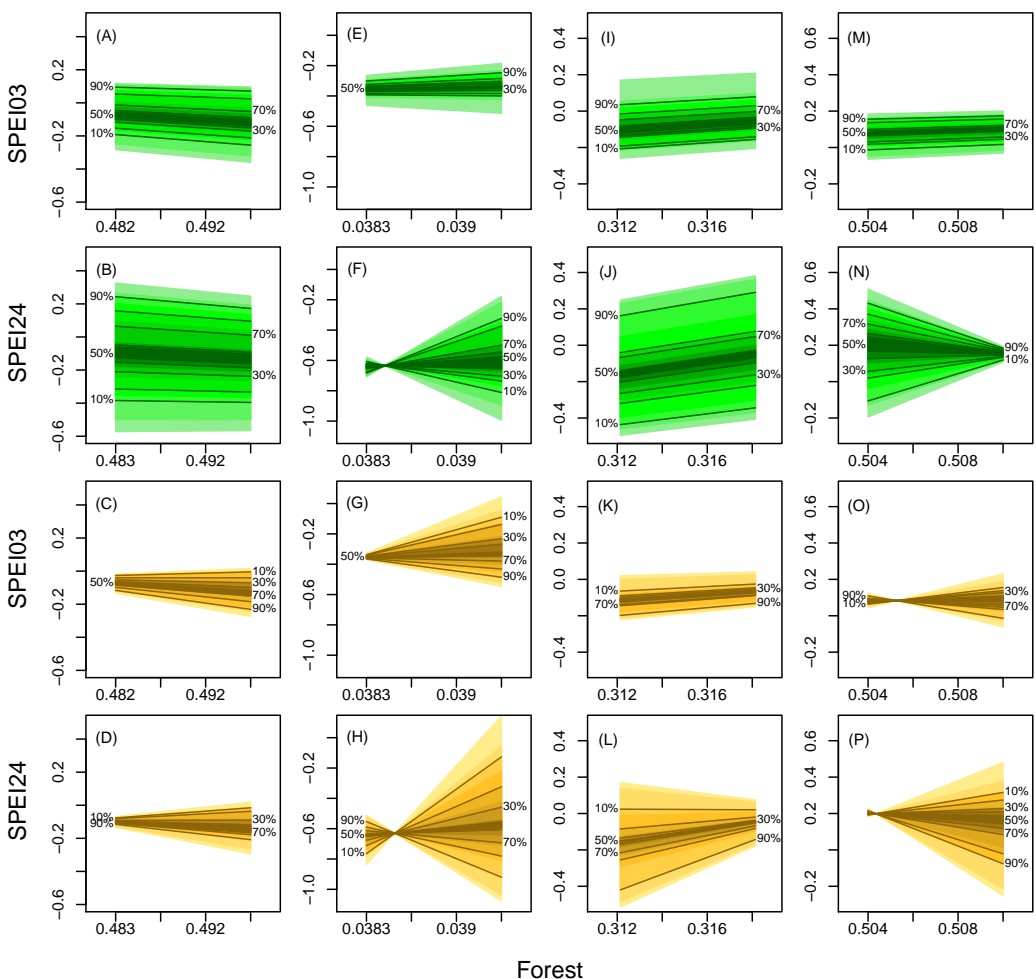

**Figure 6.** The effect of forest fraction on droughts for SPEI03 in the first and third rows; for SPEI24 in the second and fourth rows for different levels of precipitation (dark green lines) and temperature (dark yellow lines) using the model from Sec. 3 across different climate zones (from left to right: equatorial, arid, temperate, snow). For the first two rows, the temperature is held constant at its 0.5 quantiles and the forest cover effect is explored under different quantiles of precipitation. For the bottom two rows, precipitation is held constant and the effect is shown under different temperature quantiles.

This modulating effect of the forest cover effect on the drought index is a lot stronger for the arid region. For the short-term index (SPEI03, Fig. 6G), high/low temperatures lead to a notable /positive response of SPEI03 to forest cover, while the same effect is even stronger for the long-term index (SPEI24, Fig. 6H). A possible explanation is that elevated temperatures trigger greater transpiration rates from trees. When there is an abundance of trees, they collectively draw more water from the soil. This depletes the water content in the soil, and when soil moisture becomes insufficient, it results in reduced evapotranspiration.

This decrease in evapotranspiration leads to a higher vapor pressure deficit, and subsequently, an increase in $PET$. This shift toward higher $PET$ values often corresponds with a decrease in drought indices. In the presence of low temperatures, forest transpiration tends to weaken. In such conditions, the influence of forest transpiration on drought changes becomes less pronounced. It's important to note that trees can still play a role in promoting water circulation and increasing air humidity in arid regions. When there is more precipitation, especially in the presence of a substantial number of trees, it can provide some

relief from drought conditions (in the temperate region). However, the extent of this relief depends on various factors, including the amount and timing of the precipitation, the type of tree species present, and other environmental factors. For the temperate zone, increasing tree cover results in increasing rates of evaporation, which contributes to higher atmospheric moisture, thus reducing $PET$ and hence increasing SPEI. However, if temperatures are reduced, i.e. are close to their 0.1 quantiles, this effect vanishes. In the temperate region, water resources are relatively abundant compared to the arid region. This suggests that higher

temperatures and more trees can indeed increase the SPEI24, as shown in (Fig. 6L), as they contribute to higher evaporation and transpiration rates, leading to increased atmospheric moisture and potentially mitigating drought conditions. For the snow region, trees in mostly snow-covered regions change the albedo and the transpiration. Again, for the snow region, the model captures only a little part of the variability, which limits its interpretability.

The combined forest fraction and meteorological effect on drought indices vary across regions and time scales $\tau$ in mag-

nitude and direction. In equatorial and temperate regions, the long-term drought index (SPEI24) is primarily influenced by precipitation, rather than forest cover. Similarly, the short-term drought index (SPEI03) in 4 regions is also more dependent on precipitation than on forest cover. However, in the arid region, precipitation plays a significant role in modulating the influence of forest cover on the long-term drought index (SPEI24). Furthermore, temperature plays an apparent role in modulating the impact of forest cover on both short-term (SPEI03) and long-term droughts (SPEI24) in arid regions. Additionally, in the tem-

perate region, temperature also plays a substantial role in influencing the effect of forest cover on long-term drought (SPEI24) conditions.

And additional figures that provide further insights and analysis about the effect of forest cover fraction on SPEI24 and SPEI03 when temperature or precipitation is fixed at maximum or minimum levels are included in Fig. C1 and Fig. C2. The regions exhibiting more pronounced changes are primarily in the arid and snow regions. These regions experience more

significant variations in combined forest fraction and meteorological effect on drought conditions compared to other areas.

### 4.3 Response of drought to precipitation and temperature under extreme forest fraction (minimum and maximum) conditions

We studied the effect of precipitation and temperature on short-term (SPEI03) and long-term (SPEI24) drought indices for observed minimum $(\min(X_{\text{forest,region}}))$ and maximum forest cover fractions $(\max(X_{\text{forest,region}}))$. We generate a grid with $100 \times$ 100 points based on the variables of precipitation and temperature used in the linear models across different climate regions, and then, use the precipitation, temperature, and forest extremes (maximum and minimum) to calculate the drought indices (SPEI03 and SPEI24) based on the models from Eq. 6. The first two rows in Fig. 7 depict the short-term drought index (SPEI03) obtained for the minimum (first row) and maximum forest cover (second row); the last two rows depict the long-term drought index (SPEI24) for minimum (third row) and maximum forest cover (fourth row). Furthermore, similar analyses for SPEI06, SPEI12 and scPDSI can be found in Fig. C6 and Fig. C7 and Fig. C8.

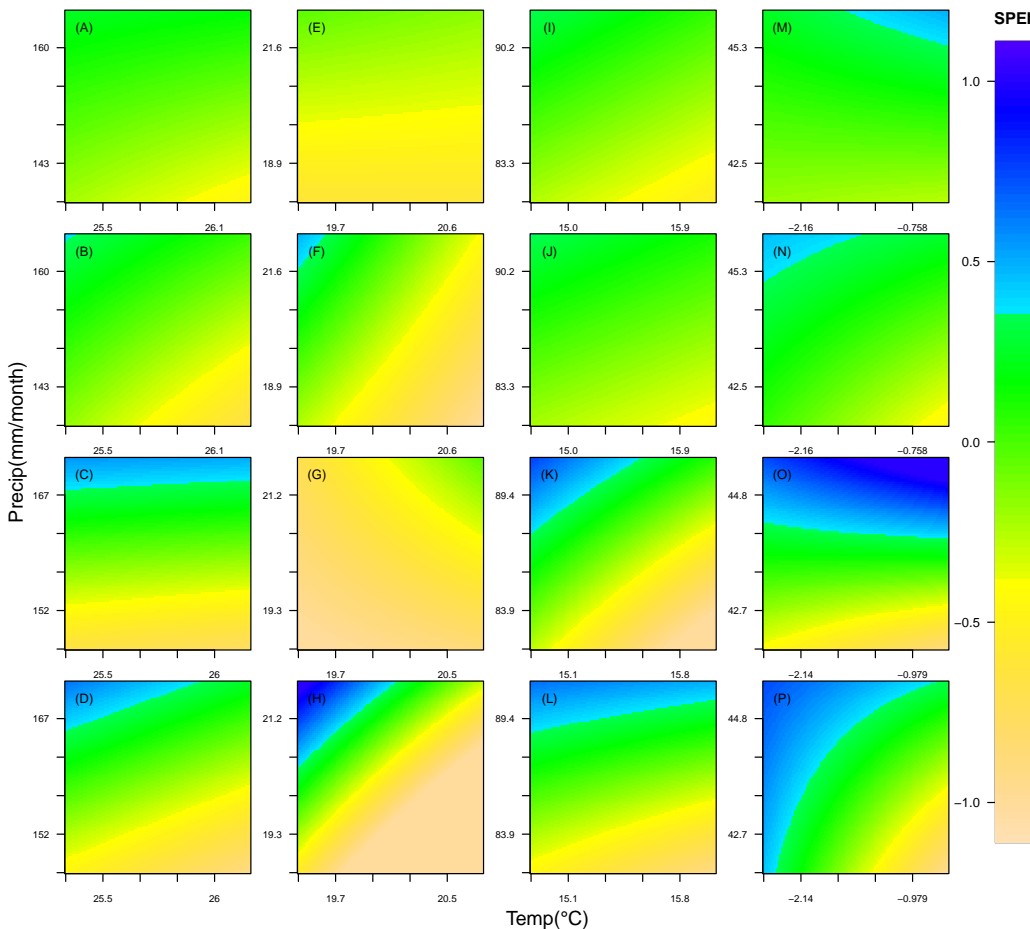

**Figure 7.** The variation of droughts (SPEI03: the first and second rows; SPEI24: the third and fourth rows) under the effect of minimal (the first and third rows) and maximal (the second and fourth rows) forest cover across four regions (from the left to right: equatorial, arid, temperate and snow regions)

Blue colors indicate the situations wetter than the median (positive indices), green represents situations close to the median (indices around 0), and yellow indicates conditions drier than the median conditions (negative indices). For the equatorial region, precipitation influences the short and long-term drought indices (vertical color change); for maximum forest cover (Fig. 7B), the dependence on temperature (horizontal color change) becomes visible, even more so for the long-term index (Fig. 7D). As the forest cover increases (Fig. 7B D), higher temperatures become more significant in driving drought conditions. The elevated temperature leads to increased rates of evaporation and transpiration, potentially making the region drier. Nevertheless, the combined effect of these factors on droughts, as indicated in Tab. B1, is found to be statistically insignificant. The long-term index in the arid region shows a somewhat stronger temperature dependence for maximum forest cover (Fig. 7H) than that in the equatorial region (Fig. 7D). For minimum forest cover (Fig. 7G), the temperature dependence is very weak

(and reversed). For SPEI03 (Fig. 7E and F), the dependence also changes from precipitation to temperature if there are more trees. In the temperate region, the SPEI24 for minimum forest cover (Fig. 7K) shows the strongest temperature and precipitation dependence; increasing forest cover to maximum (Fig. 7L) significantly reduces the temperature dependence, leaving the long-term drought index for the temperate region dominated by precipitation. The short-term index (Fig. 7I and J) is less dependent on both, and the influence from precipitation and temperature is almost unaffected by the change in forest cover.

In the snow region, the relationship between temperature, precipitation, and long-term droughts is closely linked to the extent of forest cover. The interaction between precipitation and temperature plays a crucial role in shaping drought conditions when there is maximal forest cover fraction (Fig. 7P). Minimizing the forest cover (Fig. 7O) eliminates the temperature dependence. For SPEI03 (Fig. 7M and N), the dependence on both variables is a lot weaker, with temperature dependence being reversed by reducing the forest from maximum (Fig. 7N) to the minimum (Fig. 7M).

Comparing all panels in Fig. 7, forest cover has a greater influence on the long-term drought index in the arid region. Increasing the forest cover increases the dependence on temperature in snow, arid and equatorial regions, and it reduces the dependence on temperature for the temperate region.

Fig. 5 presents that the droughts indices are greatly affected by precipitation (except arid regions). In most cases, the color shift in Fig. 7 should be vertically distributed. Transpiration from the forest is essentially the evaporation of water vapors from 440 plant leaves and stems, which is an important part of the water cycle (39% of terrestrial precipitation and 61% of evapotranspiration globally) (Schlesinger and Jasechko, 2014). within the cycle, temperature plays a major role in the rate of transpiration (Kimball, 1999), especially during the growing season. Therefore, forests can act as a medium for the temperature to have a greater impact on drought changes, as shown in snow and arid regions. In the temperate region, the drought indices are dominated by precipitation and the forest only affects the influence of temperature to SPEI24, and when there are fewer trees, the 445 influence of temperature will be amplified (shown in Fig. 6L). And this is consistent in Fig. 7K and L. Note: simultaneously with afforestation or deforestation the global climate is changing. Therefore, local changes in climate may also be additionally influenced by global climate change and not only afforestation or deforestation rates.

## 5  Conclusions

The scientific community has dedicated significant efforts to quantify the influence of land cover changes on climate. In this 450 study, linear models were employed to evaluate the impacts of forest cover and climatic factors on droughts at various temporal scales in four distinct climate regions (excluding the polar region, which has negligible forest cover). The study findings are summarized as follows:

1. Linear models incorporating forest fraction, precipitation, and temperature yield the most accurate results for explaining drought indicators in the equatorial region, but are less effective in the snow region. These three variables provide a 455 better fit for changes in SPEIs compared to scPDSI, which may be due to scPDSI's consideration of soil conditions. Precipitation is the primary factor explaining a significant proportion of the regression variance in all regions, except for the arid region where temperature is the dominant factor.

2. It is conceivable that changes in precipitation and temperature can impact the relationship between forest cover changes and drought occurrence. Specifically, precipitation alters the influence of forest cover on long-term drought (SPEI24) in the arid region, while temperature significantly modifies the effect of forest cover on both short- and long-term droughts (SPEI03 and SPEI24) in the arid region and only long-term droughts (SPEI24) in the temperate region.

3. Forest cover has differing effects on drought occurrence (especially long-term drought) depending on the maximum and minimum levels of forest fraction in different climate regions. Specifically, in arid and snow regions, higher forest cover intensifies the combined influence of precipitation and temperature for long-term drought. Conversely, in regions with lower forest cover, precipitation is the dominant factor for drought occurrence. The opposite pattern is observed in the temperate region, where lower forest cover promotes a combined effect of precipitation and temperature for long-term drought.

Despite the substantial progress made in understanding the factors that influence drought occurrence, uncertainties remain that have not always been fully acknowledged. One source of uncertainty is the use of different drought indices, which can produce divergent results. In this study, we focused on two indices, scPDSI and SPEI. However, the calculation of drought indices, particularly scPDSI, is subject to various uncertainties. For example, scPDSI requires information on temperature, precipitation, and soil conditions. Obtaining detailed soil information for each location can be challenging, leading to potential inaccuracies in the calculation of the index. In addition, human activities like irrigation are not accounted for in the index calculation, which can affect its accuracy. Meanwhile, each of the drought indices has its own niche where it excels. Different drought indices should be used when assessing different types of droughts (Mishra and Singh, 2010). Secondly, this study used a relatively short-term dataset of 25-26 years, and future studies can use longer-term data to obtain a more robust understanding of the relationship between forest cover and drought. Thirdly, our study indicates that the relative influence of precipitation and temperature on drought indices varies across different regions and forest cover fractions, however, the specific physicochemical and biological processes underlying this relationship require further verification through climate models. Finally, this study utilized linear models with a limited number of predictors, and the conclusion is valid only under the assumptions we use here. There might be other models more accurate than linear models, and future studies can investigate more complex models to explore other potential effects of forest cover change on drought. A generalization to additive models (not necessarily linear) might reveal more subtle effects. However, an initial explorative analysis with line plots did not suggest these based on the data used here. These gaps in knowledge present opportunities for future research and can help in developing a broader framework for understanding natural hazards, including droughts.

Possible extensions of this study include broadening the scope of land cover analysis to include other types of land cover, such as agricultural land, grasslands, wetlands, and settlements. Additionally, conducting more specific research in particular locations, such as Europe, could provide more definitive recommendations. In our research, to simplify the initial study, we chose to aggregate data across different climate regions. This approach helps to smooth out localized variations and complexities. Going into a more detailed spatial analysis would be a deeper level of investigation. However, it's important to note that the conclusions might not be entirely consistent when transitioning to a grid-point-wise training approach. This inconsistency

arises due to interaction terms in the model building process. Future model inter-comparison studies, such as the Land-Use Model Intercomparison Project simulations (LUMIP; Lawrence et al., 2016) under the Coupled Model Intercomparison Project phase 6 (CMIP6), could further investigate the impact of land use on climate and examine the effect of land cover change on

the onset and evolution of drought under various forcing conditions. The influence of droughts is not solely attributed to local factors such as forest cover but is also affected by global drivers and large-scale atmospheric patterns. Separating and isolating the specific effects of forest cover from these broader-scale factors presents a significant challenge in our research. Further investigation is warranted to explore the varying effects of different tree species on drought. The impact of tree species on drought dynamics can differ significantly, and thus, it is important to delve deeper into this topic. These extensions could sig-

nificantly expand our understanding of the relationship between land cover change and drought and inform the development of more effective land use policies to mitigate the impacts of climate change. Expanding our understanding of the regional and global climate impact of land cover changes, including their scale effects, can help to inform the development of land use policies that prioritize climate objectives. This is particularly important given that decisions regarding land use are frequently made at the subnational level by regional authorities.

This study enhances our comprehension of the connection between forest cover and drought across various temporal scales and climatic regions. Additionally, it elucidates the combined impact of forest cover, temperature, and precipitation on drought variability. The findings of this study can offer a theoretical framework for the creation of regional land use policies that prioritize climate concerns, as well as deepen our insight into the impact of land surface changes on climate change.

*Code and data availability.* All the datasets in the article can be downloaded from the website for free. The links have been given in Sec. 2.

The statistical analysis is finished by R. All the packages used in the paper are free and the information is given in Sec. 3.

**Appendix A: Data**

**Table A1.** Datasets needed in the study

| Varibles | Spatial resolution | download link | citation | last accessed time |
|---|---|---|---|---|
| Climate classification | 0.1 °x0.1° | https://hess.copernicus.org/articles/11/1633/2007/hess-11-1633-2007-supplement.zip | Peel et al. (2007) | Feb 2018 |
| SPEI | 0.5°x0.5° | http://digital.csic.es/handle/10261/202305 | Vicente-Serrano et al. (2010a) | March 2020 |
| scPDSI | 0.5°x0.5° | https://crudata.uea.ac.uk/cru/data/drought/ | Burke et al. (2006) | March 2021 |
| CCI land cover dataset | 300 m | http://maps.elie.ucl.ac.be/CCI/viewer/download.php | Santoro et al. (2017) | June 2021 |
| Precipitation | 0.5°x0.5° | https://crudata.uea.ac.uk/cru/data/hrg/ | (Harris et al., 2014) | March 2021 |
| Temperature | 0.5°x0.5° | https://crudata.uea.ac.uk/cru/data/hrg/ | (Harris et al., 2014) | March 2021 |

**Table A2.** The classification of SPEI values

| SPEI values | Drought classifiaction |
| --- | --- |
| 2.00 and above | Extremely wet |
| 1.50 to 1.99 | Very wet |
| 1.00 to 1.49 | Moderately wet |
| -0.99 to 0.99 | Near normal |
| -1.49 to -1.00 | Moderately dry |
| -1.99 to -1.50 | Very dry |
| -2.00 and less | Extremely dry |

**Table A3.** The classification of scPDSI values

| scPDSI values | Drought classifiaction |
| --- | --- |
| 4.00 and above | Extremely wet |
| 3.00 to 3.99 | severely wet |
| 2.00 to 2.99 | Moderately wet |
| 1.00 to 1.99 | Mildly wet |
| 0.50 to 0.99 | Incipiently wet |
| -0.49 to 0.49 | Near normal |
| -0.99 to -0.50 | Incipiently dry |
| -1.99 to -1.00 | Mildly dry |
| -2.99 to -2.00 | Moderately dry |
| -3.99 to -3.00 | severely dry |
| -4.00 and less | Extremely dry |

**Table B1.** Significance from two-sided $t$-tests for coefficients being compatible with 0 in the linear models across different climate zones (equatorial, arid, temperate, snow) and for the scPDSI and SPEI$\tau$ with different integration times. (level of significance '***' 0.001, '**' 0.01, '*' 0.05, '.' 0.1)

| | | $X_{\text{forest}}$ | $X_{\text{precip}}$ | $X_{\text{temp}}$ | $X_{\text{forest}} : X_{\text{precip}}$ | $X_{\text{forest}} : X_{\text{temp}}$ | $X_{\text{precip}} : X_{\text{temp}}$ |
|---|---|---|---|---|---|---|---|
| | scPDSI | | *** | *** | | | |
| | SPEI03 | | *** | *** | | | |
| Equatorial | SPEI06 | | *** | *** | | | |
| | SPEI12 | | *** | * | | | |
| | SPEI24 | | *** | * | | | |
| | scPDSI | ** | *** | | | | |
| | SPEI03 | | ** | ** | | * | |
| Arid | SPEI06 | | ** | ** | | * | |
| | SPEI12 | | *** | *** | * | ** | |
| | SPEI24 | | ** | ** | * | ** | |
| | scPDSI | * | *** | * | . | * | * |
| | SPEI03 | | *** | ** | | | |
| Temperate | SPEI06 | | *** | *** | | | |
| | SPEI12 | | *** | *** | | | |
| | SPEI24 | . | *** | *** | | . | |
| | scPDSI | | * | | | | |
| | SPEI03 | | ** | | | | |
| Snow | SPEI06 | | *** | | | | |
| | SPEI12 | | | ** | . | | |
| | SPEI24 | | * | * | | | |

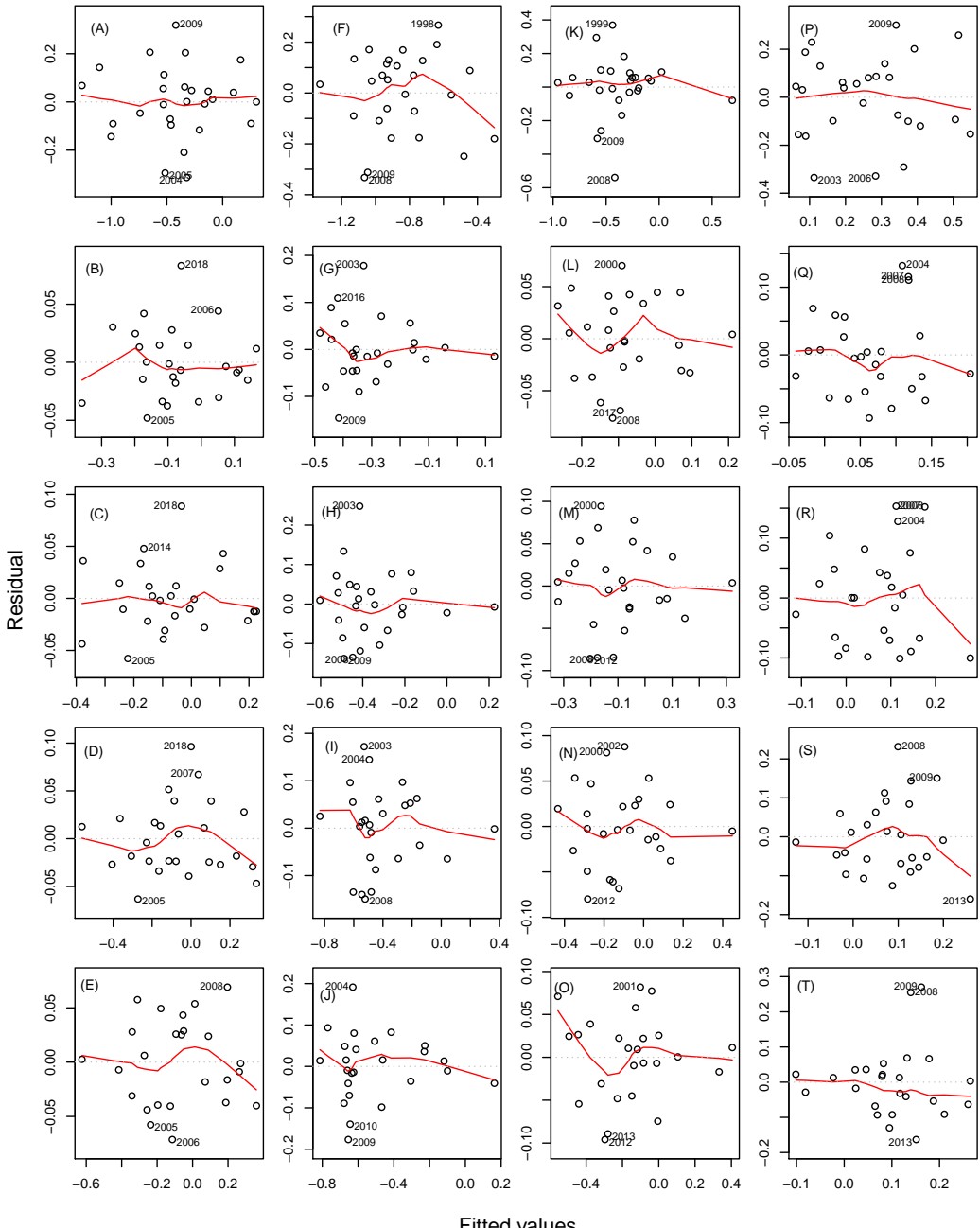

**Figure B1.** Residuals vs fitted for the linear models across different climate zones (columns from left to right: equatorial, arid, temperate, snow) and for the scPDSI and SPEI$\tau$ with different integration times (rows from top to bottom: scPDSI and SPEI$\tau$ with $\tau \in \{03, 06, 12, 24\}$).

# Appendix C: Results

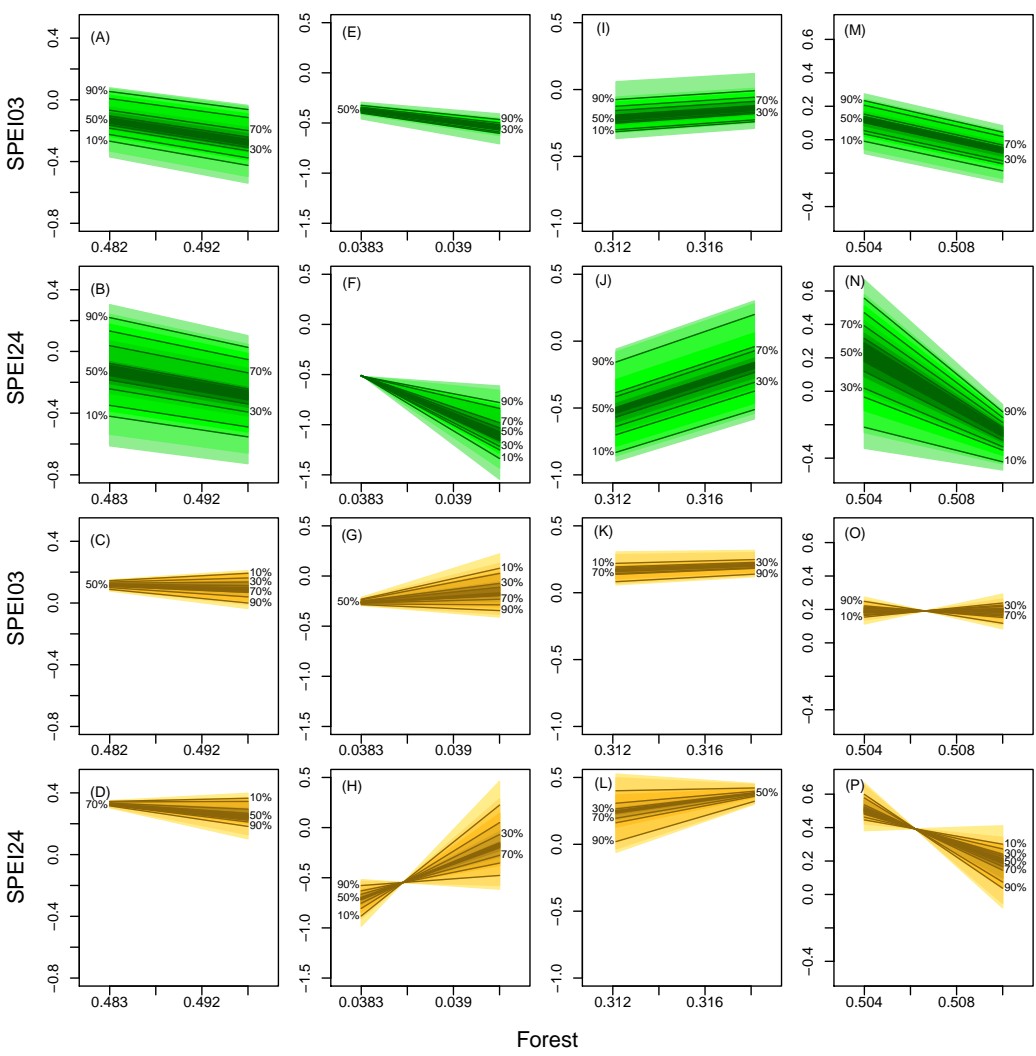

**Figure C1.** Same as Fig. 6 but for the first two rows, the temperature is held constant at its 0.99 quantiles (maximum) and for the bottom two rows, precipitation is held constant at its 0.99 quantiles(maximum).

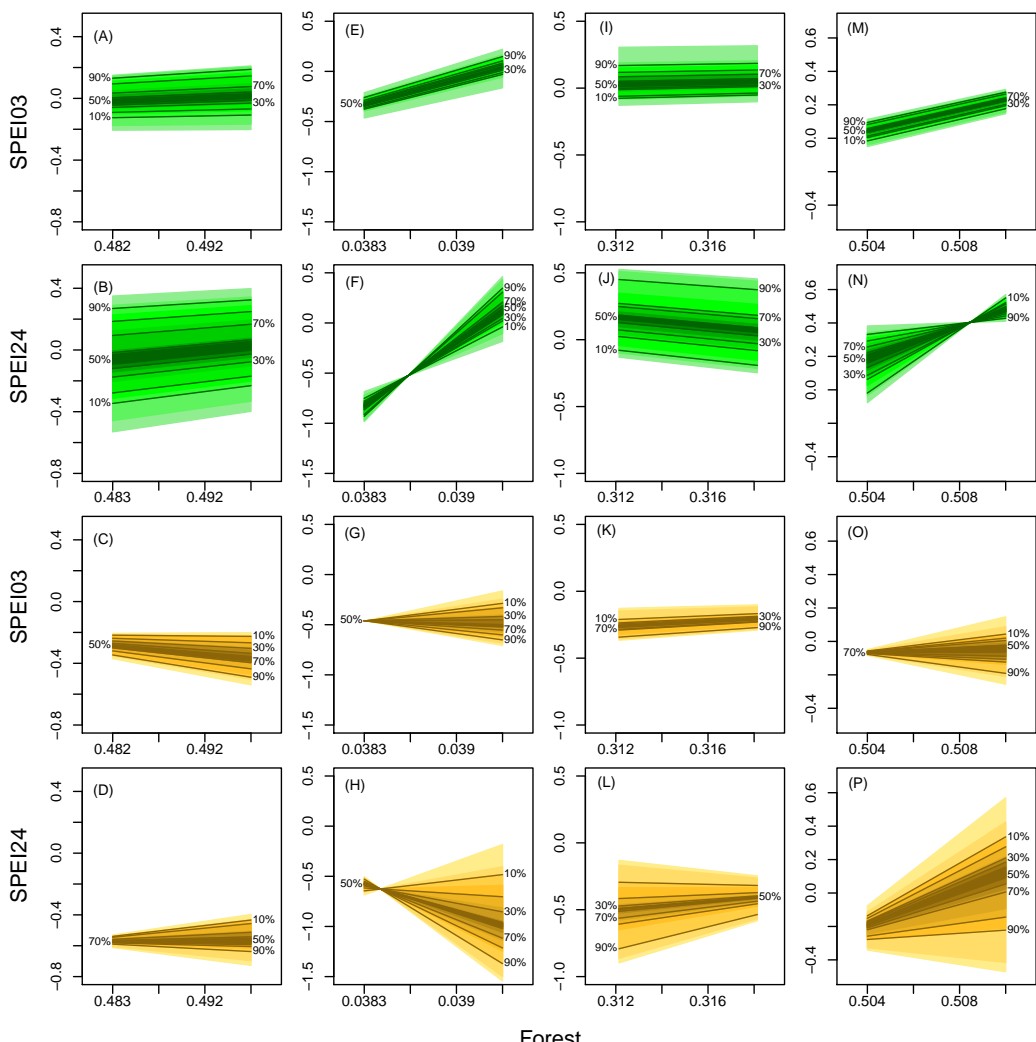

**Figure C2.** Same as Fig. 6 but for the first two rows, the temperature is held constant at its 0.01 quantiles (minimum) and for the bottom two rows, precipitation is held constant at its 0.01 quantiles (minimum).

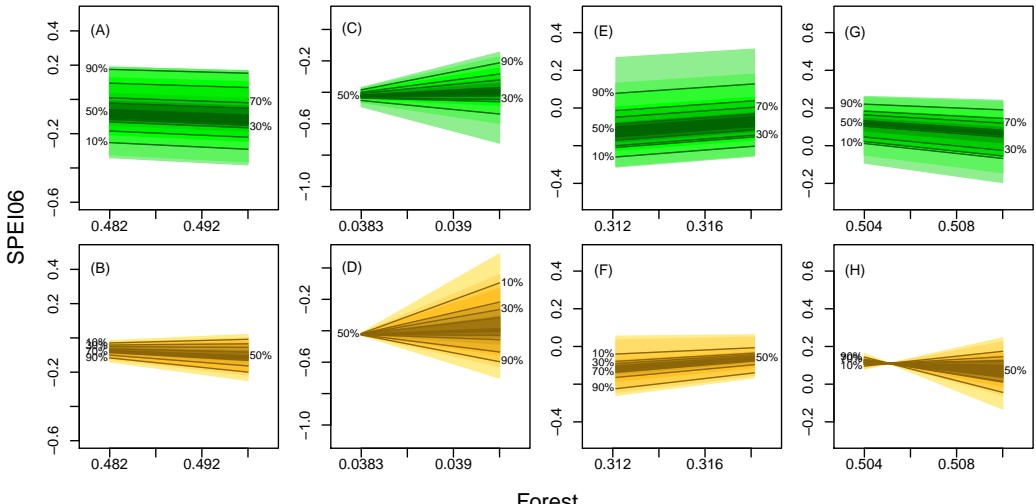

**Figure C3.** Same as Fig. 6 but for SPEI06

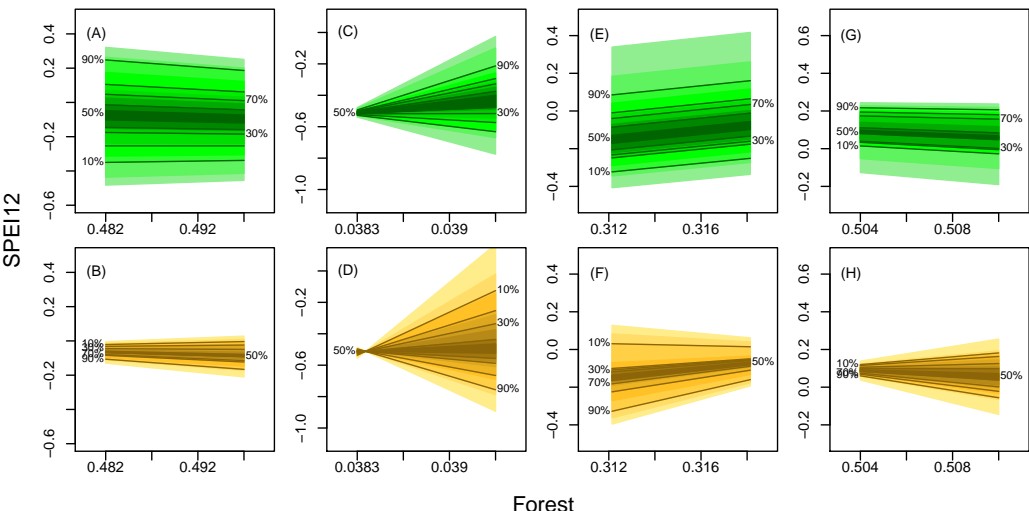

**Figure C4.** Same as Fig. 6 but for SPEI12.

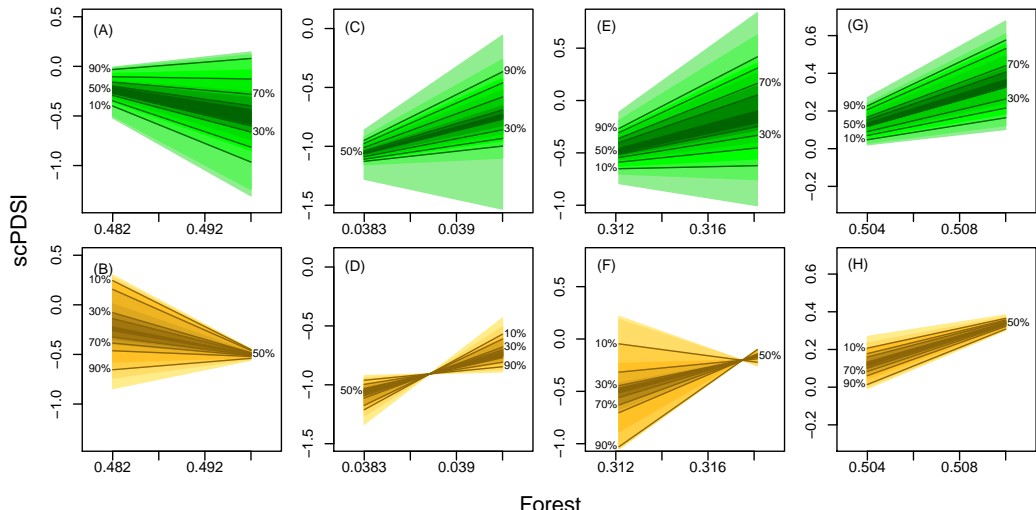

**Figure C5.** Same as Fig. 6 but for scPDSI.

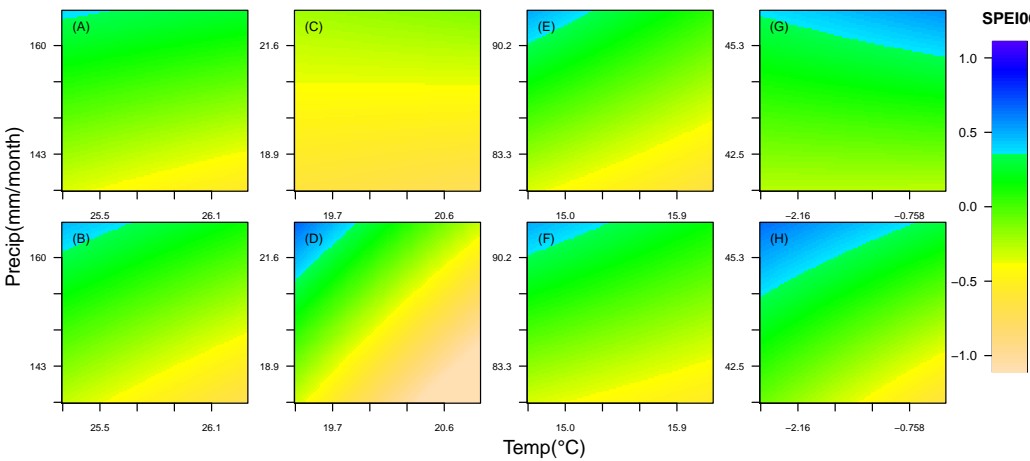

**Figure C6.** Same as Fig. 7 but for SPEI06.

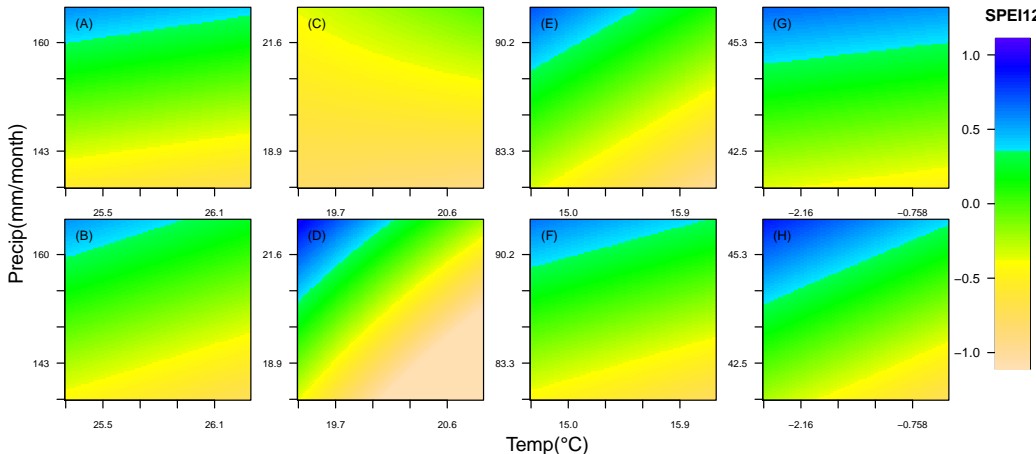

**Figure C7.** Same as Fig. 7 but for SPEI12.

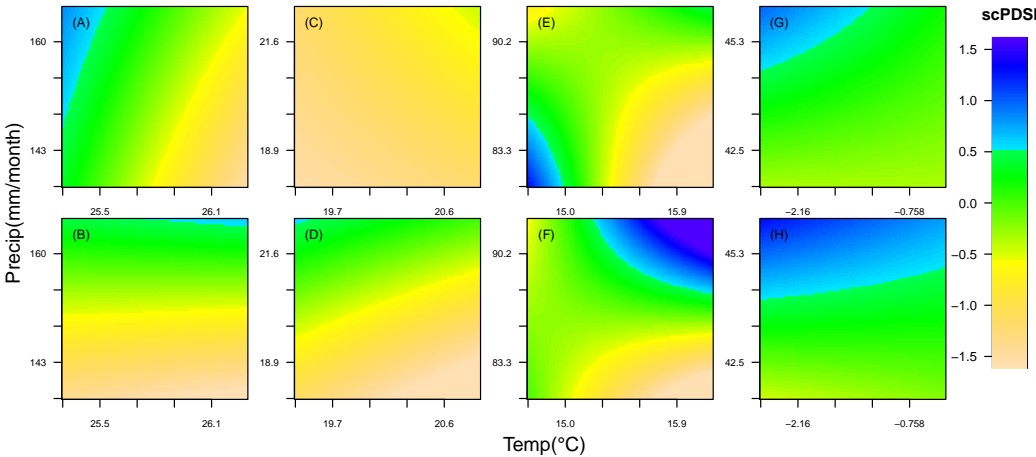

**Figure C8.** Same as Fig. 7 but for scPDSI.

*Author contributions.* YL and HWR conceived and designed the research. YL built the models with the primary processing data from BH, and conducted the statistical analysis under the supervision of HWR. YL made the figures. All authors interpreted the results and wrote the paper.


*Competing interests.* The authors declare that they have no conflict of interest.

*Acknowledgements.* YL acknowledges the support from the China Scholarship Council (CSC). BH acknowledges the support of the Norwegian Research Council (project no. 294534 and 286773).

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
