# Peer review of "Using statistical models to depict the response of multi-time scales drought to forest cover change across climate zones"

_Hydrology and Earth System Sciences, 2023_

## Referee Comment (RC1)

*Review of* **"Using statistical models to depict the response of multi-time scales drought to forest cover change across climate zones" by** *Yan Li, et al., 2023*

The authors use linear models to explore the influence of forest cover, temperature, and precipitation on the drought indices in various climate zones. The study's motivation and goal are exciting for the community. The exploratory data analysis used in this study is robust and could be interpreted very well. However, I have a few major comments, which shall be clarified/discussed further:

- How do you isolate the local effects of forest cover and drought from the global drivers and large-scale atmospheric patterns? For example, increase/decrease in precipitation, anthropogenic global warming, jet-stream shift, ITCZ, etc.? The tree growth dependency on T and P depends on the biomes (Boreal forest, Temperate seasonal forest, etc.). Each tree has its characteristics.

- It should be described why the authors used linear models for their analysis.

- One suggestion which might be considered to add value to the results:
Using the linear regression model is an excellent approach to analysing the interactions between the variables and features. However, as mentioned by the authors, the interplay among precipitation, temperature, soil, land cover and drought might be complex and non-linear. Authors could add some complexity to the model by using decision-tree-based models already implemented in R and comparing the results with the linear model. On the other hand, simple/shallow decision tree models are also interpretable.

**Other comments:**

**Lines 1-5:** What do you mean by forest cover change? Do you mean human-made changes or natural changes?

**Line 6:** Hard to understand: ""to explore the changes in forest fraction and drought from 1992–2018."". Do you mean to find a kind of relationship between those two? Or exploring them separately? And why those 27 years?

**Line 7:** which various factors? Please clarify! Are they natural factors or management factors, etc. ..

**Lines 8-9:** Is precipitation the dominant one among the two variables? Please mention!

**Lines 9-10:** It needs to be clarified: You mention precipitation and temperature (which describe the climate state), then forest cover and finally, short and long-term drought. The reader needs to catch up

on the clear goal. Please clarify which relationships or driving effects you will explore in this manuscript.
Some chains like: T, P => forest cover => drought?

**Line 30**: "–500 million hectares up to +1000 million hectares" what do -500 million hectares mean? And all the SSPs show the same trend, or do they differ from each other?

**Lines 35-45:** maybe you could also mention that extensive forests like the Amazon are the sink of CO2 and are predicted to become a source of CO2 under the recent trend of climate change we are following:

Boulton, C.A., Lenton, T.M. & Boers, N. **Pronounced loss of Amazon rainforest resilience since the early 2000s.** *Nat. Clim. Chang.* **12**, 271–278 (2022). https://doi.org/10.1038/s41558-022-01287-8

**Line 57-58:** "in this region," which region mention again.

**Line 59-60:** Is drought a condition or a phenomenon? Clarify? There are also many definitions for drought, like meteorological, agricultural, etc.… Please clarify how you define the drought. Which index do you use? Is it based on temperature and precipitation, or other variables, like soil moisture, evaporation, etc. are, involved?

**Line 62-63:** What do you mean by "forest structure and carbon content"? Please clarify.

**Line 65:** How much increase in the frequency and intensity? Is it significant? With respect to which period? Please describe in more detail!

**Lines 79-80:** The word ""change"" is used frequently.

**Lines 87-88:** Why didn't you use the newer version of the data with a higher resolution or cite this study:

Beck, H., Zimmermann, N., McVicar, T. *et al.* Present and future Köppen-Geiger climate classification maps at 1-km resolution. *Sci Data* **5**, 180214 (2018). https://doi.org/10.1038/sdata.2018.214

**Figure 1:** Please insert the number of grid points belonging to each main climate classification.

**Line 116:** Does the potential evapotranspiration data have a reference?

Please include a table with the characteristics of the data used in this study to have a better overview. For example, it is boring to know when and where you downloaded each dataset. A table would be enough, which describes all the datasets. And please include the citation of each dataset in the table.

**Figure.2.** Given that the scPDSI values between -1 and 1 are considered normal, how significant are the annual trends shown in the drought indices in Fig.2?

**Line 172:** Why not consider the precipitation sum (yearsum) instead of mean (yearmean)?

**Line 175:** What do you mean by complex? Clarify!

**Lines 200-215:** How about the problem of collinearity? There might be correlations between the forest cover change and P or T. How do you consider this? A correlation matrix might show the collinearity between the predictor variables, or the Variance inflation factors (VIF) method might help. The other concern is how many grid points you achieve for each climate zone. How big is the training dataset for each climate zone? I assume you have a more extensive training dataset for the temperate than the equatorial zone. How about the seasonality? You have an arid zone in both the North and South hemisphere. Averaging over all those grid points might mix seasons. Could one include the latitude as an extra feature in the lm model?

Equation 6: Is "i" indicating the observation over different grid points and times? Or do you average the gridpoints of each climate zone at each time, and "i" is just the time? Do you train for each grid point a separate linear model? Do you train one linear model for each climate zone? Please clarify in more detail….

**Line 261:** Must be moved to data and methods.

**Line 262:** You mentioned the regions before.

**Lines 283-284:** How does the time deviation of forest cover look like in equatorial regions? There may be some temporal changes in tropical forest cover. This is because the trees receive enough energy (T) and moisture (P) throughout the year. Have you removed the seasonal cycle from the "lm" features, i.e., T, P and forest?

**Lines 295-297:** It is a strong conclusion based on a single linear statistical model. I would be cautious about concluding solid results on this.

**Figure 6:** I see green and yellow colours and not blue and red lines. Using symbols instead of colours could help readers with colour blindness.

**Lines 364-366:** Given that the trees' species might change in the snow and equatorial regions, how do you isolate those impacts?

**368-369:** I am unsure if this is the correct English: "The colour change … should be vertical". Please re-frame.

**Line 382-383:** Please mention that your conclusion is valid only under the assumptions you use here. There might be other models more accurate than your linear model.

**Line 414:** You have to spell out CMIP.

---

## Referee Comment (RC2)

Forest cover changes (e.g., deforestation and afforestation) have profound impacts on climate through biophysical processes. Prior studies have mostly focused on the impacts of forest cover changes on temperature and precipitation. However, the impacts of forest cover changes on drought have not been sufficiently examined and remain largely unknown. In this paper, Li and the coauthors try to fill this knowledge gap using a statistical model. They found varying effects of forest cover changes on different time-scale droughts across climate zones. Moreover, the impacts of forest cover changes may vary with precipitation and temperature within a climate zone.

This paper is well organized, clearly written and presents some novel results on the impacts of forest cover change on drought. However, I am a little concerned about whether the statistical model used in this work is a useful tool to address the relevant questions. Moreover, the statistical model-based results are not sufficiently convincing due to the lack of mechanisms or explanations in some cases. I think that the methods and results should be further clarified or explained. Please see my specific comments listed below.

Major:

1. The authors used a statistical model, and the model is in principle a linear multiple regression model. While the model can reasonably reproduce the year-to-year variations in drought in equatorial, arid and temperate regions, it is difficult for us to interpret the results and mechanisms derived from such a statistical model. For example, changes in drought can be attributed to changes in forest cover,

precipitation and temperature and the interactions between the three variables in a mathematical way (Equation 5). However, how can the individual effects on drought be interpreted? Specifically, what does the effect of forest and precipitation interactions on drought ($X_{forest}$: $X_{precip}$) mean? Does $X_{forest}$: $X_{precip}$ mean that precipitation changes influence forest cover and subsequently drought or forest cover changes influence precipitation and subsequently drought?

2. Owing to the shortcomings of the statistical model mentioned above, some results based on the statistical model are also not clearly explained. For example, Figure 6F shows that SPEI24 decreases as forest cover increases when precipitation is low. The authors explain that a small amount of water is transpired into the atmosphere due to a high fraction of available trees (Line 310-311). Here are two problems. First, why does a higher tree cover fraction contribute to lower evapotranspiration when precipitation is low? Second, how are changes in evapotranspiration further related to changes in drought? Moreover, Figure 6L shows that the SPEI24 increases as forest cover increases when precipitation is high. The authors explain that "the types of trees here can adapt their leaves and roots to absorb all of the excess water (Line 328-329)", but they do not explicitly explain the positive response of SPEI to forest cover changes. Furthermore, some results shown in Figure 7 are not sufficiently explained. For example, it remains unclear why the dependence of drought on temperature and precipitation varies with forest cover.

3. I note that the forest cover range (X-axis) in Figure 6 varies with region, but why?

In arid regions, forest cover ranges between 0.0383 and 0.0393 (Figure 6L), and such a range (~ 0.001) is much smaller than the historical actual changes in forest cover (Figure 2). Why? Such a small increase in forest cover even corresponds to an increase of 0.3 in SPEI24 when precipitation is high (Figure 6L). It can be estimated that historical actual loess in forest cover (~ -3) will cause a decrease of 900 in SPEI24 in arid regions.

4. The authors categorize the global land into four climate zones and aggregate the forest cover, precipitation and temperature values within a climate zone for further analysis. I can not understand why the author do this. Forest cover, precipitation and temperature are spatially highly heterogeneous within a climate zone. Therefore, why not apply the statistical model pixel by pixel?

5. The authors selected temperature, precipitation and forest cover as three independent variables to build the statistical model (Equation 5). An implicit assumption is that the authors think that temperature, precipitation and forest cover can largely explain the annual variation in drought, but why? I do not doubt the contribution of temperature and precipitation to the evaluation of drought. However, it remains unclear why the other human activities (e.g., aerosol emission) are not considered here. It is also feasible to either replace the forest cover change with other human forcing or combine the forest cover change with other human forcing to rebuild the statistical model. It is unclear how the main results shown in this manuscript would be modified if different independent variables are selected to

build the model.

6. From Figure 2 and Figure 3, I see that the drought indices show a clear decreasing trend in arid regions during the analysis period. I'm curious about whether such a trend is related to global warming. If so, this is not surprising to see a dominant contribution of temperature to the evolution of drought, as shown in Figure 5. In other words, the covariance of drought is dominated by its long-term trend, which is further related to the long-term temperature trend in arid regions. This leads to another question: whether the drought indices, temperature and precipitation need to be detrended before regression? The authors do not detrend the variables and may confound the contribution of temperature, precipitation and forest cover changes to drought at multi-time scales.

7. In Figure 6, the authors show the responses of drought indices to forest cover changes at different precipitation (or temperature) levels with temperature (or precipitation) fixed at its median. It is unclear whether the main results would be modified if temperature or precipitation is fixed at other levels (e.g., maximum or minimum).

8. In Figure 6 and 7, the authors only show the results for SPEI03 and SPEI24. It is fine to only show these two drought indices in the main text, but the results for the other indices (i.e., SPEI06, SPEI12 and scPDSI) should be provided, for example, in the supplementary material.

Minor:

1. Line 6: "forest fraction" -> "forest cover fraction".

2. Line 9: "The impact of forest cover" -> "The impact of forest cover changes".

3. Line 10-12: "forest cover's impact" -> "the impact of forest cover changes"

4. Line 38-39: "forests typically have a low surface albedo" -> "the typically low surface albedo of forests".

5. Line 39-42: I think that the large uncertainty in the temperature effect of afforestation/deforestation in the mid-latitude is MAINLY caused by the radiative (i.e., albedo) and nonradiative (i.e., roughness and evapotranspiration) effects being similar in magnitude but opposite in sign. The background climate, forest types or analysis methods, as mentioned by the authors, just further enlarge such an uncertainty.

6. Line 47-58: In this paragraph, the authors review the impacts of deforestation/afforestation on precipitation in previous studies. I find that most references cited here are either old (before 2010) or review articles (e.g., Bonan, 2008; Perugini et al., 2017). Numerous important studies have examined the impacts of deforestation/afforestation on precipitation based on observations (Leite-Filho et al. 2021; Smith et al. 2023) and simulations (Liang et al. 2022; Luo et al. 2022) in recent years. I recommend the authors to update the references in this

paragraph.

7. Line 60: "And it is" -> "Drought is".

8. Line 106-107: "…, which maps…". What does "which" refer to? SPEI or SPI? Rephrase this sentence. When I first read this sentence, I interpreted "which" as "SPEI". As such, I cannot understand why the authors say that the SPEI use precipitation as the only input but later they say that potential evapotranspiration is also used. I later realized that "which" refers to "SPI".

9. Section 2.2: In this section, you should tell the readers what the magnitude and sign of the SPEI and scPDSI mean. For example, what are the possible ranges of the indices? What do the positive or negative values of the indices mean? What do higher or lower values of the indices mean?

10. Figure 3: The description of figure caption is inaccurate. It should be the annual means of precipitation and temperature aggregated analogously to the aggregation level of the drought index, rather than the annual temperature and precipitation.

11. Line 193-194: Why not considering the interactions between $X_1$ and $X_2$ (i.e., $X_1:X_2$) and $X_1$ and $X_3$ (i.e., $X_1:X_3$)? Do you assume that $X_1$ is independent of $X_2$ and $X_3$?

12. Line 199: Why are the forth and fifth right-hand terms are the same in Equation 5?

13. Line 200-201: What do the annual mean precipitation (i.e., $X_{precip}$) and temperature (i.e., $X_{temp}$) refer to? Do they refer to the commonly used mean values of

precipitation and temperature or the mean values of the precipitation and temperature aggregated to the aggregation level of the drought index (as mentioned in Line 172)? Clarify the "annual mean" here. Does Dτ also refer to the annual mean values of scPDSI or SPEI?

14. Line 228: What does $\hat{y}_i$ denote?

15. Line 275: "ominates" might be "dominates"?

16. Line 320-321: High/low temperatures lead to a notable negative/positive response of SPEI03 to forest cover, instead of decrease/increase in forest cover.

References:

Leite-Filho, A.T., et al. 2021: Deforestation reduces rainfall and agricultural revenues in the Brazilian Amazon. *Nature Communications*, 12, 2591.

Liang, Y., et al. 2022: Deforestation drives desiccation in global monsoon region. *Earth's Future*, 10, e2022EF002863.

Luo, X., et al. 2022: The biophysical impacts of deforestation on precipitation: results from the CMIP6 model intercomparison. *Journal of Climate*, 35(11), 3293-3311.

Smith, C., et al. 2023: Tropical deforestation causes large reductions in observed precipitation. *Nature*, 615, 270–275.

---

## Author Response (AR1)

Authors' Response to Reviews of

**Using statistical models to depict the response of multi-time scales drought to forest cover change across climate zones**

*Yan Li, Bo Huang and Henning W. Rust*

*Hydrology and Earth System Sciences (HESS)*

RC: Reviewers' Comment    EC: Editor' Comment    AR: Authors' Response

Dear Prof. Dr. Genevieve Ali,

We would like to express our gratitude for the opportunity to submit a revised version of our manuscript entitled "Using statistical models to depict the response of multi-time scales drought to forest cover change across climate zones". We sincerely appreciate the time and effort invested by both you and the reviewers in providing valuable feedback on our work. The insightful comments and suggestions provided by the reviewers have greatly contributed to improving the quality of our manuscript. We have carefully considered their feedback and made the necessary changes accordingly. Throughout the revised manuscript, we have highlighted the modifications we have made in response to their recommendations.

Generally, in the supplementary information, we have included three tables and nine figures in response to the reviewers' comments. Supplementary Table A1 provides a summary of the datasets used in the analysis. Table A2 presents the classification of the SPEI drought index, while Table A3 displays the classification of the scPDSI index. Significance tests for the p-values obtained from the linear regression models conducted in various climate zones, as well as for the scPDSI and SPEI$\tau$ with different integration times, are presented in Table B1. Figure B1 illustrates the residuals vs fitted values for the linear models across different climate zones. Additionally, Figures C1-C8 depict the supplementary analysis for different drought indices across various climate zones.

Please do not hesitate to contact me should you need any more information,

Yours sincerely,

Yan Li
(On behalf of the co-authors)

**Editor**

EC: While I understand and agree with you that linear models are "easy/straightforward" to apply, the reviewers are both mentioning, as their first question/comment, that they have a bit of an issue with that methodological choice. Hence, in your response to reviewers as well as in the manuscript, I think that a strong/stronger argument should be made for why the use of a linear model is realistic, from a physical/process-based standpoint.

AR: We include in the text "We use linear models because of their great flexibility, versatility and robustness. They are characterized by linearity in parameters to estimate; relations between predictand and variables in the predictor can still be formulated in a non-linear way. Furthermore, they easily allow to describe joint effects of different variables (temperature and forest cover) on the predictand (interactions), a feature made extensive use of in this study." in Line 215-218. In (generalized) linear models, it is possible to incorporate interactions into the modeling process. The notation and framework for (generalized) linear models were initially introduced by McCullagh and Nelder in 1989. We explain this in Line 234-237. And according to Table 1, MSE (Mean Squared Error) and Adjusted $R^2$ ($R^2_{adj}$) for all models, these three factors exhibit a certain power in describing drought changes. Hence, linear models comprising these factors are well-suited to describe the drought indices in the equatorial, temperate, and arid regions, while the snow region requires more intricate considerations when examining the factors impacting drought indices. Furthermore, we have included residual versus fits plots in Figure B1. Upon careful inspection of the residual plots, no evident structures or patterns are observed, indicating that there are no apparent missing terms or heteroscedasticity present in the models.

Considering all factors and considerations, the selection of a linear model is deemed suitable for this study. Taking into account the specific research objectives, available data, and the nature of the relationships being investigated, a linear model provides a relevant and appropriate framework for analyzing the variables under investigation. By utilizing a linear model, we can effectively explore and quantify the relationships between the variables in a straightforward and interpretable manner. As we also assume relations to be at least smooth (if not linear), we suggest using additive models for further studies. And we add the text "A generalization to additive models (not necessary linear) might reveal more subtle effects. However, an initial explorative analysis with line plots did not suggest these based on the data used here." in Line 475-477.

EC: In response to some reviewer comments, you have provided explanations but did not indicate whether a shorter version of those explanations/clarifications would be included in the revised manuscript as well. I assume so, but I still wanted to mention it to guide your revision process.

AR: In response to the majority of the questions and comments, we have provided explanations within the revised manuscript itself. We have ensured to indicate the specific line numbers where the corresponding explanations have been added.

**Anonymous Referee #1**

RC: The authors use linear models to explore the influence of forest cover, temperature, and precipitation on the drought indices in various climate zones. The study's motivation and goal are exciting for the community. The exploratory data analysis used in this study is robust and could be interpreted very well. However, I have a few major comments, which shall be clarified/discussed further:

AR: Thanks for the appreciation of our work and the useful feedbacks to improve the quality of the manuscript.

RC: How do you isolate the local effects of forest cover and drought from the global drivers and large- scale atmospheric patterns? For example, increase/decrease in precipitation, anthropogenic global warming, jet-stream shift, ITCZ, etc.? The tree growth dependency on T and P depends on the biomes (Boreal forest, Temperate seasonal forest, etc.). Each tree has its characteristics.

AR: Isolating the local effects of forest cover and drought from global drivers and large-scale atmospheric patterns is a challenging task. It is difficult to completely separate the effects of global drivers, such as anthropogenic global warming and jet-stream shifts, from local effects. We employed several statistical models in this study to explore the potential impact of forest cover change on drought across different climate zones. However, it is important to note that statistical models can only provide an estimation of the true relationship between forest cover change and drought change. The coefficient of determination (Adjusted $R^2$) of the statistical models varied between 0.23 to 0.97 (as shown in Table 1), indicating that these models can only partially explain the relationship between forest cover change and drought change. In future studies, it is crucial to consider multiple factors and utilize a variety of methods, such as statistical models, remote sensing, and ground-based measurements, to obtain a more comprehensive understanding of the dynamics between forest cover and drought.

Regarding tree growth, it is true that tree growth can depend on temperature and precipitation, and the specific characteristics of each tree species. However, in this study, we focus on the overall effect of forest cover change on drought, rather than the specific effect of each tree species. We use a satellite-based forest cover dataset, which provides information on the extent of forest cover across different regions, rather than information on the specific tree species present in each region.

And thank you for your comments, which is excellent extension for our research. The text "The influence of droughts is not solely attributed to local factors such as forest cover but is also affected by global drivers and large-scale atmospheric patterns. Separating and isolating the specific effects of forest cover from these broader-scale factors presents a significant challenge in our research. Further investigation is warranted to explore the varying effects of different tree species on drought. The impact of tree species on drought dynamics can differ significantly, and thus, it is important to delve deeper into this topic." are included in Line 484-488.

RC: It should be described why the authors used linear models for their analysis.

AR: We include in the text "We use linear models because of their great flexibility, versatility and robustness. They are characterized by linearity in parameters to estimate; relations between predictand and variables in the predictor can still be formulated in a non-linear way. Furthermore, they easily allow to describe joint effects of different variables (temperature and

forest cover) on the predictand (interactions), a feature made extensive use of in this study." in Line 215-218.

RC: One suggestion which might be considered to add value to the results: Using the linear regression model is an excellent approach to analysing the interactions between the variables and features. However, as mentioned by the authors, the interplay among precipitation, temperature, soil, land cover and drought might be complex and non-linear. Authors could add some complexity to the model by using decision-tree-based models already implemented in R and comparing the results with the linear model. On the other hand, simple/shallow decision tree models are also interpretable.

AR: Indeed, a regression tree model could be an interesting extension to our approach. However, we did not see any indication for the need of more complex models based on explorative line plots analysis made prior to building the regression models. Furthermore, a further splitting of data — which a tree-based model brings analog — would probably lead to less robust results. As we also assume relations to be at least smooth (if not linear), we suggest using additive models for further studies. And we add the text "A generalization to additive models (not necessary linear) might reveal more subtle effects. However, an initial explorative analysis with line plots did not suggest these based on the data used here." in Line 475-477.

Other comments:

RC: Lines 1-5: What do you mean by forest cover change? Do you mean human-made changes or natural changes?

AR: In this study, we are not making a distinction between whether the changes in forest cover are a result of human activities or natural processes. We are solely examining the impact of changes in forest area on drought.

RC: Line 6: Hard to understand: ""to explore the changes in forest fraction and drought from 1992–2018."". Do you mean to find a kind of relationship between those two? Or exploring them separately? And why those 27 years?

AR: Our aim is to investigate the impact of changes in forest area on drought across different timescales and climate zones. To achieve this, we begin by examining the average changes in forest cover and drought indices across four distinct climate zones. Subsequently, we construct a series of statistical models to identify the relationship between changes in forest area and drought. We changed the word "changes" Line 6 to "relationship". For our analysis, we utilize the ESA CCI land cover data, which was available for the period of 1992-2018 at the time of our study.

RC: Line 7: which various factors? Please clarify! Are they natural factors or management factors, etc.

AR: In this study, the term "various factors" refers to forest cover change, precipitation, and temperature, which we clarified in Line 7. It is important to note that we only considered forest area changes based on the ESA CCI map and did not distinguish between changes caused by natural factors or management practices.

RC: Lines 8-9: Is precipitation the dominant one among the two variables? Please mention!

AR: The impact of forest cover change, precipitation, and temperature on the change of droughts varies across different regions. Precipitation appears to be the dominant factor (among the tree factors) affecting the change of droughts in the equatorial, temperate, and snow regions, whereas in the arid region, temperature is the dominant factor among the three. It is clarified in the manuscript (Line 8-9).

RC: Lines 9-10: It needs to be clarified: You mention precipitation and temperature (which describe the climate state), then forest cover and finally, short and long-term drought. The reader needs to catch up on the clear goal. Please clarify which relationships or driving effects you will explore in this manuscript.
Some chains like: T, P => forest cover => drought?

AR: In this study, we aimed to investigate the influence of forest cover change and meteorological factors, such as precipitation and temperature, on droughts at different time scales. In Section 4.1, we analyzed the effects of forest cover change and meteorological factors on droughts based on Analysis of Variance mentioned in Section 3.2. In Section 4.2, we focused on how meteorological factors influence the impact of forest cover change on droughts. Finally, in Section 4.3, we examined the effects of meteorological factors (precipitation and temperature) on droughts under extreme values of forest cover area, specifically the maximum and minimum values. The expression has been clarified in the revised manuscript (Line 104-108).

RC: Line 30: "−500 million hectares up to +1000 million hectares" what do -500 million hectares mean? And all the SSPs show the same trend, or do they differ from each other?

AR: Popp et al. (2017) reported a systematic understanding of the Shared Socio-Economic Pathways (SSPs) and their potential impacts on land-use changes, agriculture, food security, greenhouse gas emissions, and related issues. The authors employed five Integrated Assessment Models, each equipped with distinctive land-use modules, to convert the SSP narratives into quantitative projections. We give details in text "Based on five integrated assessment models and Shared Socioeconomic Pathways (SSP) scenarios, global forest areas are likely to decrease up to -600 million hectares in SSP3 (regional rivalry) and increase by up to 1100 million hectares in SSP1 (sustainability) by the end of the 21st century" to explain the future forest changes under SSP scenarios (Line 31-34).

RC: Lines 35-45: maybe you could also mention that extensive forests like the Amazon are the sink of CO2 and are predicted to become a source of CO2 under the recent trend of climate change we are following:

Boulton, C.A., Lenton, T.M. & Boers, N. Pronounced loss of Amazon rainforest resilience since the early 2000s. Nat. Clim. Chang. 12, 271–278 (2022). https://doi.org/10.1038/s41558-022-01287-8

AR: We add this reference in the new version (Line 24-26).

RC: Line 57-58: "in this region," which region mention again.

AR: Clarified. The region is temperate region (Line 70).

RC: Line 59-60: Is drought a condition or a phenomenon? Clarify? There are also many definitions for drought, like meteorological, agricultural, etc.... Please clarify how you define the drought. Which index do you use? Is it based on temperature and precipitation, or other variables, like soil moisture, evaporation, etc. are, involved?

AR: Our primary focus is on the changes in drought conditions. It should be noted that there are various definitions of drought, including meteorological drought, agricultural drought, hydrological drought, and others. For our study, we use two specific indices to measure drought: The Standardized Precipitation Evapotranspiration Index (SPEI) and the self-calibrated Palmer Drought Severity Index (scPDSI). The SPEI is based on the difference between precipitation and potential evapotranspiration, assuming a Gaussian distribution. And the scPDSI is based on a two-layer soil model, incorporating not only meteorological variables but also soil condition factors, making it an appropriate index for measuring hydrological drought. The information is give in details in Section 2.2. And we also changed the word "phenomenon" to "condition" in Line 76.

RC: Line 62-63: What do you mean by "forest structure and carbon content"? Please clarify.

AR: Changes in drought patterns can affect the forest structure by altering the growth and survival of different species, leading to changes in biodiversity and ecosystem functioning. In addition, changes in forest structure can affect carbon storage by altering the distribution and types of vegetation that store carbon. Meanwhile, drought can lead to increased mortality and decreased growth of trees, which can reduce the amount of carbon stored in the forest ecosystem. We added the explanation in the new manuscript (Line 78-79).

RC: Line 65: How much increase in the frequency and intensity? Is it significant? With respect to which period? Please describe in more detail!

AR: The scope of this study was limited to the analysis of the changes in drought conditions (dry or wet), and did not include an investigation of the frequency change and intensity. Further research can explore the impact of forest cover area change on the frequency and intensity of drought across different climate zones. And we added the explanation in details in the new manuscript (Line 81-88).

RC: Lines 79-80: The word ""change"" is used frequently.

AR: Revised in Line 100-101.

RC: Lines 87-88: Why didn't you use the newer version of the data with a higher resolution or cite this study:

Beck, H., Zimmermann, N., McVicar, T. et al. Present and future Köppen-Geiger climate classification maps at 1-km resolution. Sci Data 5, 180214 (2018). https://doi.org/10.1038/sdata.2018.214

AR: Several versions of the Köppen-Geiger climate classification maps have been generated over time. In this study, we utilized the main climate zones from the Köppen-Geiger climate classification system. We compared the main climate zones in the latest version and the version we used, and found no significant differences. Therefore, we decided to keep the current climate zone classification and provided an explanation for it in the revised manuscript (Line 118-121).

RC: Figure 1: Please insert the number of grid points belonging to each main climate classification.

AR: We added the grid points for the five main climate zones in the new version (Line 122-123).

RC: Line 116: Does the potential evapotranspiration data have a reference?

Please include a table with the characteristics of the data used in this study to have a better overview. For example, it is boring to know when and where you downloaded each dataset. A table would be enough, which describes all the datasets. And please include the citation of each dataset in the table.

AR: The potential evapotranspiration in this study was calculated using the FAO-56 Penman-Monteith method, and the necessary datasets for the calculation were obtained from the CRU TS3.24.01 dataset. More details regarding the data used in this study can be found in the supplementary document, specifically in Table A1.

RC: Figure.2. Given that the scPDSI values between -1 and 1 are considered normal, how significant are the annual trends shown in the drought indices in Fig.2?

AR: In Figure 2, we present the scPDSI values that have been averaged for each region and year. So these values are more towards the average. According to Dai (2013), the scPDSI values ranging from -0.5 to -1.0, -1.0 to -2.0, -2.0 to -3.0, and -3.0 to -4.0 correspond to dry spell, mild drought, moderate drought, and severe drought, respectively. By analyzing the changes in scPDSI values, we can infer the trend of dryness or wetness. And we also add Table A3 to show the classification of scPDSI in details.

RC: Line 172: Why not consider the precipitation sum (yearsum) instead of mean (yearmean)?

AR: Since we use standardized precipitation, temperature, and forest cover area to build our linear models, it does not matter whether we use precipitation sum (yearsum) or mean (yearmean) in our analysis.

RC: Line 175: What do you mean by complex? Clarify!

AR: Based on the observations from Figure 3 P, Q, R, S, T, we cannot establish a clear relationship between the drought indices and precipitation or temperature in the snow region. This implies that precipitation or temperature may not be the dominant factors in this region. Other factors such as their interaction or other environmental variables may play a more important role in driving the changes in drought conditions in the region (Line 205-207).

RC: Lines 200-215: How about the problem of collinearity? There might be correlations between the forest cover change and P or T. How do you consider this? A correlation matrix might show the collinearity between the predictor variables, or the Variance inflation factors (VIF) method might help. The other concern is how many grid points you achieve for each climate zone. How big is the training dataset for each climate zone? I assume you have a more extensive training dataset for the temperate than the equatorial zone. How about the seasonality? You have an arid zone in both the North and South hemisphere. Averaging over all those grid points might mix seasons. Could one include the latitude as an extra feature in the lm model?

AR: In order to ensure the accuracy of our linear models, we also examined the collinearity among the independent variables in Equation 5 using the Variance Inflation Factor (VIF) function in R. All VIFs were found to be less than 5, indicating moderate correlations between the variables but not severe enough to require attention (added in Line 246-248).

It should be noted that for the equatorial region, there were 11,030 points; for the arid region, there were 15,673 points; for the temperate region, there were 9,587 points; and for the snow region, there were 20,734 points. However, prior to model construction, all the necessary data were averaged both regionally and annually. In each linear model, the data length was 25 or 26 years since we only analyzed the annual change of various variables and filtered out the seasonal influence. As shown in Fig. 1, the climate classification is also latitude-related, so if we use different linear models for different climate regions, we do not need to add latitude as a variable in model building.

RC: Equation 6: Is "i" indicating the observation over different grid points and times? Or do you average the grid points of each climate zone at each time, and "i" is just the time? Do you train for each grid point a separate linear model? Do you train one linear model for each climate zone? Please clarify in more detail....

AR: For each region and variable, we averaged the grid points, resulting in a time series of 25 or 26 years with "i" representing the number of observation years. We utilized forest cover, precipitation, temperature, and drought indices (on different time scales) datasets to train a linear model for each region. Therefore, the study resulted in 20 linear models (4 climate regions and 5 drought indices: scPDSI and SPEIs). In Section 4.1, we analyzed the contribution of different factors to drought across different time scales and climate regions. In Sections 4.2 and 4.3, we conducted sensitive experiments based on the linear models to assess the interaction of forest cover and meteorological factors to droughts across different time scales and climate regions. Relevant information regarding this is presented in Line 146-148, Line 160-161, and Line 177-179.

RC: Line 261: Must be moved to data and methods.

AR: The sentence has been deleted in the section.

RC: Line 262: You mentioned the regions before.

AR: Deleted.

RC: Lines 283-284: How does the time deviation of forest cover look like in equatorial regions? There may be some temporal changes in tropical forest cover. This is because the trees receive enough energy (T) and moisture (P) throughout the year. Have you removed the seasonal cycle from the "lm" features, i.e., T, P and forest?

AR: The variable (forest cover) we used in the models are annual value, so it has filtered out the seasonal influences.

RC: Lines 295-297: It is a strong conclusion based on a single linear statistical model. I would be cautious about concluding solid results on this.

AR: Yes. The conclusions presented in this study are based on a set of linear statistical models with a limited number of predictors. It is important to note that this approach has its limitations

and future studies can investigate more complex models to explore other potential effects of forest cover change on drought. However, we chose to use linear models because they are easy to implement, interpret and efficient to train. We would like to emphasize that this study is not perfect, but it can still be meaningful if it helps people to better understand the complex relationship between land use and climate change. We add some sentences to discuss it. (Line 475-477).

RC: Figure 6: I see green and yellow colours and not blue and red lines. Using symbols instead of colours could help readers with colour blindness.

AR: Modifying the figure from colour lines to symbols may not be feasible. However, we clarify the colours used in the figure by explicitly stating that dark green and dark yellow represent the effect of precipitation and temperature, respectively.

RC: Lines 364-366: Given that the trees' species might change in the snow and equatorial regions, how do you isolate those impacts?

AR: Our current study only focuses on the forest cover area change, and we are not explicitly accounting for the potential impacts of changes in tree species on drought conditions. However, changes in tree species could potentially affect the water cycle and ecosystem functioning in ways that could indirectly impact drought conditions. In order to isolate these impacts, more detailed studies would be needed that incorporate information on the specific tree species and their water use characteristics in different regions. This information could then be used to refine models and better understand the complex relationships between forest cover, tree species, and drought conditions in different ecosystems. It can be done in the future research. And we include the text "The impact of tree species on drought dynamics can differ significantly, and thus, it is important to delve deeper into this topic." in Line 487-488.

RC: Line 368-369: I am unsure if this is the correct English: "The colour change ... should be vertical". Please re-frame.

AR: Clarified (Line 431-432).

RC: Line 382-383: Please mention that your conclusion is valid only under the assumptions you use here. There might be other models more accurate than your linear model.

AR: We add some sentences to clarify the limitation and possible application of our conclusion. (Line 472-475)

RC: Line 414: You have to spell out CMIP.

AR: The full name of CMIP is given in the main text (Line 482-483).

Reference:

Dai, A, 2013: Increasing drought under global warming in observations and models. *Nature Climate Change*, **3**, 52.

Popp, A., and Coauthors, 2017: Land-use futures in the shared socio-economic pathways. *Global Environmental Change*, **42,** 331-345.

**Anonymous Referee #2**

RC: Forest cover changes (e.g., deforestation and afforestation) have profound impacts on climate through biophysical processes. Prior studies have mostly focused on the impacts of forest cover changes on temperature and precipitation. However, the impacts of forest cover changes on drought have not been sufficiently examined and remain largely unknown. In this paper, Li and the coauthors try to fill this knowledge gap using a statistical model. They found varying effects of forest cover changes on different timescale droughts across climate zones. Moreover, the impacts of forest cover changes may vary with precipitation and temperature within a climate zone.

This paper is well organized, clearly written and presents some novel results on the impacts of forest cover change on drought. However, I am a little concerned about whether the statistical model used in this work is a useful tool to address the relevant questions. Moreover, the statistical model-based results are not sufficiently convincing due to the lack of mechanisms or explanations in some cases. I think that the methods and results should be further clarified or explained. Please see my specific comments listed below.

AR: Thanks for the appreciation of our work and the useful feedbacks to improve the quality of the manuscript.

Major:

1. The authors used a statistical model, and the model is in principle a linear multiple regression model. While the model can reasonably reproduce the year-to-year variations in drought in equatorial, arid and temperate regions, it is difficult for us to interpret the results and mechanisms derived from such a statistical model. For example, changes in drought can be attributed to changes in forest cover, precipitation and temperature and the interactions between the three variables in a mathematical way (Equation 5). However, how can the individual effects on drought be interpreted? Specifically, what does the effect of forest and precipitation interactions on drought (Xforest: Xprecip) mean? Does Xforest: Xprecip mean that precipitation changes influence forest cover and subsequently drought or forest cover changes influence precipitation and subsequently drought?

AR: We adopt the notation for (generalized) linear models as introduced by McCullagh and Nelder (1989). While interpreting so-called direct effects response ~ term is relatively straightforward, understanding interactions can be more challenging. To facilitate intuitive interpretation, we propose the following approach: consider a simple model with a direct effect, where response ~ term_A. Mathematically, this is represented as $Y = a_0 + a_1 X_1 + \varepsilon$, with $a_0$ and $a_1$ as unknown but constant parameters. If $a_1$ is found to be dependent on another term, $X_2$, we can model $a_1$ itself linearly, resulting in $Y = a_0 + a_1 X_1 + \varepsilon = a_0 + (b_0 + b_1 X_2)X_1 + \varepsilon$, where $b_0$ and $b_1$ are again unknown but constant parameters. The effect of $X_1$ on Y now depends on the value of $X_2$. Expanding further, we have $Y = a_0 + b_0 X_1 + b_1 X_2 X_1 + \varepsilon$. In model notation, we can express this as $Y \sim X_1 + X_2 : X_1$. The interpretation of $X_2 : X_1$ is a modulation of the effect of $X_1$ on Y by $X_2$ (or vice versa: modulation of the effect of $X_2$ on Y by $X_1$). Another perspective is to view this as approximating the unknown function $Y = f(X_1, X_2)$ using a second-order Taylor expansion, with the resulting unknown parameters estimated from data. By employing this approach to investigate how meteorological conditions

and forest cover influence droughts, we aim to generate ideas for potential mechanisms based on data. we also add new text in Line 234-237 to explain the interaction in the linear model.

RC: 2. Owing to the shortcomings of the statistical model mentioned above, some results based on the statistical model are also not clearly explained. For example, Figure 6F shows that SPEI24 decreases as forest cover increases when precipitation is low. The authors explain that a small amount of water is transpired into the atmosphere due to a high fraction of available trees (Line 310-311). Here are two problems. First, why does a higher tree cover fraction contribute to lower evapotranspiration when precipitation is low? Second, how are changes in evapotranspiration further related to changes in drought? Moreover, Figure 6L shows that the SPEI24 increases as forest cover increases when precipitation is high. The authors explain that "the types of trees here can adapt their leaves and roots to absorb all of the excess water (Line 328-329)", but they do not explicitly explain the positive response of SPEI to forest cover changes. Furthermore, some results shown in Figure 7 are not sufficiently explained. For example, it remains unclear why the dependence of drought on temperature and precipitation varies with forest cover.

AR: Here, we give more explanations.

SPEI = P-PET with Potential Evapotranspiration (PET) being an upper bound of what could be transpired if there was sufficient water supply and sufficient vegetation to realize evapotranspiration. More trees will transpire more water but this does not directly affect PET, as PET is only the potential evapotranspiration. However, there are indirect effects mediated by the effect of trees on temperature and vapour pressure deficit (humidity). The latter two variables in turn influence PET. As we evaluate the model for constant temperatures (in Figure 6F), humidity is the only remaining factor influencing PET and thus SPEI in the following way: increasing vapour pressure deficit (drier air) leads to increasing PET and thus decreasing SPEI. For larger amounts of precipitation, we expect more transpiration to be realized with increasing forest fraction leading to a smaller vapour pressure deficit and thus also reduced PET; Hence increasing values for the SPEI with increasing forest fraction. If precipitation is lower, this effect decreases and the slopes in Figure 6F get smaller. There is not sufficient water to be evapotranspirated, even if the forest fraction increases. For a specific amount of precipitation (about the median) the slope is 0. For less than this amount, we see a negative slope suggesting the interpretation that for restricted water supply, an increase of trees leads to an increase of PET and hence to a decrease of SPEI. As we do not see a potential explanatory mechanism for this effect, we expect that it is due to the restriction of the model to a linear change in slope with precipitation and extrapolation beyond the validity of the model. This case calls for a more detailed investigation allowing for a non-linear change of slope with precipitation. We include the new explanation in Line 356-361.

It's important to note that trees can still play a role in promoting water circulation and increasing air humidity in arid regions. When there is more precipitation, especially in the presence of a substantial number of trees, it can provide some relief from drought conditions (in the temperate region). However, the extent of this relief depends on various factors, including the amount and timing of the precipitation, the type of tree species present, and other environmental factors.

In Figure 6L, we aim to depict the impact of changes in forest cover on SPEI24 in the temperate region, with temperature as a conditioning factor and precipitation fixed at its 0.5 quantiles. Increasing tree cover results in increasing rates of evaporation, which contributes to higher atmospheric moisture, thus reducing PET and hence increasing SPEI. However, if temperatures are reduced, i.e. are close to their 0.1 quantiles, this effect vanishes. In the temperate region, water resources are relatively abundant compared to the arid region. This suggests that higher

temperatures and more trees can indeed increase the SPEI24, as shown in Figure 6L, as they contribute to higher evaporation and transpiration rates, leading to increased atmospheric moisture and potentially mitigating drought conditions. We include the new explanation in Line 375-384.

Figure 7 aims to investigate the impact of precipitation and temperature on drought indices while considering the observed minimum and maximum forest cover fractions in different regions. In the equatorial region, drought indices are primarily influenced by changes in precipitation when there is less tree cover. However, as the forest cover increases (row 2 and 4), higher temperatures become more significant in driving drought conditions. The elevated temperature leads to increased rates of evaporation and transpiration, potentially making the region drier. Therefore, in regions with comparable water availability, forests can act as a medium for the temperature to have a greater impact on drought changes. In arid regions, as more trees are present, the influence of temperature becomes more visible as it shapes drought conditions in the way that for constant precipitation increasing temperature leads to increasing SPEI. In temperate regions, SPEI24 is strongly influenced by temperature in the minimum-forest-cover plot (7K), this effect is reduced in the maximum-forest-cover plot (7L), consistent with Figure 6L. For given precipitation, we see an increase of SPEI from Figure 7K (minimum forest) to Figure 7L (maximum forest) for high temperatures but not so much for low temperatures. Increasing forest cover reduced the dependence of SPEI on temperature. We hypothesize that increasing forest cover leads to more transpiration and thus less water vapour deficit in the air which implies reduced PET and thus increased SPEI. When the forest fraction is maximal, the influence of precipitation becomes more visible, as demonstrated in Figure 7. In snow regions, the interaction between precipitation and temperature plays a crucial role in shaping drought conditions when there is a maximal forest cover fraction (Figure 7P). However, when the forest cover fraction is minimal, precipitation remains the dominant factor affecting drought changes for SPEI24 (Figure 7O). Thus increasing forest cover in regions with sufficient water supply by precipitation implies an increase of transpiration with increasing temperature and thus a reduction of PET, leading to an increase of SPEI. We include the new explanation in Line 413-415 and 424-425.

Overall, our study indicates that the relative influence of precipitation and temperature on drought indices varies across different regions and forest cover fractions, however, the specific physicochemical and biological processes underlying this relationship require further verification through climate models. And we also add the text in Line 470-472.

The significance tests for all linear models have been incorporated into Table B1 to help us better understand the influence in this part.

RC: 3. I note that the forest cover range (X-axis) in Figure 6 varies with region, but why? In arid regions, forest cover ranges between 0.0383 and 0.0393 (Figure 6L), and such a range (~ 0.001) is much smaller than the historical actual changes in forest cover (Figure 2). Why? Such a small increase in forest cover even corresponds to an increase of 0.3 in SPEI24 when precipitation is high (Figure 6L). It can be estimated that historical actual loess in forest cover (~ -3) will cause a decrease of 900 in SPEI24 in arid regions.

AR: In Figure 6, the X-axis represents the forest cover fraction in various regions. The graph illustrates that the equatorial and snow regions have approximately 50% forest cover fraction, while the arid region has a relatively lower forest cover fraction. Despite the arid region having less forest cover, the results obtained from the linear models indicate that this region is particularly susceptible to the impacts of forest cover change. This finding suggests that even a small change in forest cover can have a significant influence on drought conditions in the arid

region. It is important to note that the forest cover values depicted in Figure 2 have undergone a standardization process, where they have been centered and scaled to unit variance. Therefore, the forest cover fractions presented in Figure 6 and Figure 2 are not directly comparable since they are not represented in the same unit. The scaling and standardization of the forest cover values were done to facilitate analysis and interpretation within the context of the specific study.

4. The authors categorize the global land into four climate zones and aggregate the forest cover, precipitation and temperature values within a climate zone for further analysis. I cannot understand why the author do this. Forest cover, precipitation and temperature are spatially highly heterogeneous within a climate zone. Therefore, why not apply the statistical model pixel by pixel?

AR: Thanks for the remark! A pixel-based analysis would of course be possible but also a lot more complex. To reduce the complexity of an initial study, we decided to integrate over climate regions in order to average out various localized effects. Going into more detail, also spatially, would be further analysis and beyond the scope of this work.

5. The authors selected temperature, precipitation and forest cover as three independent variables to build the statistical model (Equation 5). An implicit assumption is that the authors think that temperature, precipitation and forest cover can largely explain the annual variation in drought, but why? I do not doubt the contribution of temperature and precipitation to the evaluation of drought. However, it remains unclear why the other human activities (e.g., aerosol emission) are not considered here. It is also feasible to either replace the forest cover change with other human forcing or combine the forest cover change with other human forcing to rebuild the statistical model. It is unclear how the main results shown in this manuscript would be modified if different independent variables are selected to build the model.

AR: Our primary focus is to analyse the impact of forest cover change on droughts, considering both natural and human-induced alterations. We do not differentiate between these two sources of change and instead focus on the overall change in the forest area. It is widely acknowledged that precipitation and temperature are key factors influencing drought conditions. Consequently, we aim to investigate the interactions between forest cover, precipitation, and temperature to gain a deeper understanding of their combined effects on droughts.

According to Table 1, these three factors exhibit a certain power in describing drought changes. From visual inspection and comparison of $R^2_{adj}$, drought indices in the equatorial region can be described best ($0.84 < R^2_{adj} < 0.97$). However, in the snow region, the model's performance is comparatively weaker ($0.23 < R^2_{adj} < 0.39$), suggesting that factors influencing drought indices in this region are likely more complex than the linear models considered in this study. In arid and temperate zones, the linear models incorporating forest cover, temperature, and precipitation yield results that are nearly as effective as those in the equatorial zone. Hence, linear models comprising these factors are well-suited to describe the drought indices in the equatorial, temperate, and arid regions, while the snow region requires more intricate considerations when examining the factors impacting drought indices.

6. From Figure 2 and Figure 3, I see that the drought indices show a clear decreasing trend in arid regions during the analysis period. I'm curious about whether such a trend is related to global warming. If so, this is not surprising to see a dominant contribution of temperature to the evolution of drought, as shown in Figure 5. In other words, the covariance of drought is dominated by its long-term trend, which is further related to the long-term temperature trend in arid regions. This leads to another question: whether the drought indices, temperature and

precipitation need to be detrended before regression? The authors do not detrend the variables and may confound the contribution of temperature, precipitation and forest cover changes to drought at multi-time scales.

AR: Thanks for this remark. Detrending might be indeed meaningful for some analyses. However, we do not consider it purposeful as we focus on a statistical description of drought indices as functions of temperature, precipitation and forest cover, independent on external drivers of these variables. We agree that the observed trend in drought indices may very well be a result of global warming but we expect it to be mediated by changes in regional patterns of precipitation and temperature, which we have both in our model. We do not expect a large effect of global warming on droughts which is not mediated via temperature and precipitation. Furthermore, we have included residual versus fits plots for all drought indices (scPDSI, SPEI03, SPEI06, SPEI12, SPEI24) across different regions (equatorial, arid, temperate, snow regions) in Figure B1. These plots help us identify any underlying trends over time that may not be accounted for by the existing predictor variables (temperature and precipitation). If there were missing terms in the predictor or if heteroscedasticity (unequal variance) existed, it would likely manifest as noticeable patterns or structures in the residual plots. However, upon careful inspection of the residual plots, no evident structures or patterns are observed, indicating that there are no apparent missing terms or heteroscedasticity present in the model. Therefore, there is no extra trend. We add the related explanation in Line 287-291.

7. In Figure 6, the authors show the responses of drought indices to forest cover changes at different precipitation (or temperature) levels with temperature (or precipitation) fixed at its median. It is unclear whether the main results would be modified if temperature or precipitation is fixed at other levels (e.g., maximum or minimum).

AR: Thank you for your asking here. In the 4.2 section of the study, the objective is to examine how the influence of forest cover change in droughts is conditioned by precipitation and temperature. In Figure 6, the first two rows depict scenarios where the temperature is fixed at its median value. The purpose is to observe how variations in precipitation affect the relationship between forest cover change and droughts for "normal" (i.e. median) temperature. In the last two rows of the figure, precipitation is held at a normal level, allowing for an examination of the interaction between temperature and forest cover change in relation to droughts. In Figure 7 we can observe the effect of simultaneous extremes of precipitation and temperature for two extreme cases of forest cover. For complex relations, presentation on 2D plots is always a compromise. And additional figures that provide further insights and analysis when temperature or precipitation is fixed at maximum or minimum levels have been included in Figure C1 and C2, and some explanation in Line 395-398.

8. In Figure 6 and 7, the authors only show the results for SPEI03 and SPEI24. It is fine to only show these two drought indices in the main text, but the results for the other indices (i.e., SPEI06, SPEI12 and scPDSI) should be provided, for example, in the supplementary material.

AR: Thank you for your advice. We have added the new figures in the Appendix part (Figure C3-C8).

Minor:

1. Line 6: "forest fraction" -> "forest cover fraction".

AR: modified in Line 6.

2. Line 9: "The impact of forest cover" -> "The impact of forest cover changes".

AR: modified in Line 10.

3. Line 10-12: "forest cover's impact" -> "the impact of forest cover changes"

AR: modified in Line 11.

4. Line 38-39: "forests typically have a low surface albedo" -> "the typically low surface albedo of forests".

AR: modified in Line 41.

5. Line 39-42: I think that the large uncertainty in the temperature effect of afforestation/deforestation in the mid-latitude is MAINLY caused by the radiative (i.e., albedo) and nonradiative (i.e., roughness and evapotranspiration) effects being similar in magnitude but opposite in sign. The background climate, forest types or analysis methods, as mentioned by the authors, just further enlarge such an uncertainty.

AR: Thank you for your advice. I have reorganised this part. Rvised in Line 43-47 of the new manuscript.

6. Line 47-58: In this paragraph, the authors review the impacts of deforestation/afforestation on precipitation in previous studies. I find that most references cited here are either old (before 2010) or review articles (e.g., Bonan, 2008; Perugini et al., 2017). Numerous important studies have examined the impacts of deforestation/afforestation on precipitation based on observations (LeiteFilho et al. 2021; Smith et al. 2023) and simulations (Liang et al. 2022; Luo et al. 2022) in recent years. I recommend the authors to update the references in this paragraph.

AR: We sincerely appreciate the valuable comments. We have updated the references in the new version of the manuscript in Line 55-63.

7. Line 60: "And it is" -> "Drought is".

AR: modified in Line 76.

8. Line 106-107: "…, which maps…". What does "which" refer to? SPEI or SPI? Rephrase this sentence. When I first read this sentence, I interpreted "which" as "SPEI". As such, I cannot understand why the authors say that the SPEI use precipitation as the only input but later they say that potential evapotranspiration is also used. I later realized that "which" refers to "SPI".

AR: Revised in Line 133-135. By incorporating Equation 1 in Line 138-140, we aim to enhance clarity and provide a more accessible explanation of the SPEI definition.

9. Section 2.2: In this section, you should tell the readers what the magnitude and sign of the SPEI and scPDSI mean. For example, what are the possible ranges of the indices? What do the positive or negative values of the indices mean? What do higher or lower values of the indices mean?

AR: The explanation has been added in Line 162-167. And more details in Table A2 and Table A3. we present the scPDSI and SPEIs values that have been averaged for each region and year. So these values are more towards the average.

10. Figure 3: The description of figure caption is inaccurate. It should be the annual means of precipitation and temperature aggregated analogously to the aggregation level of the drought index, rather than the annual temperature and precipitation.

AR: Modified in Figure 3.

11. Line 193-194: Why not considering the interactions between X1 and X2 (i.e., X1:X2) and X1 and X3 (i.e., X1:X3)? Do you assume that X1 is independent of X2 and X3?

AR: Here is just an example to explain the direct effect factors $(X_1, X_2, X_3)$ and the interaction of 2 factors $(X_2, X_3)$. And the total effect of $X_2 \wedge X_3$ means $X_2 * X_3$, which means the $X_2 + X_3 + X_2 : X_3$, the direct effect and interaction of the two factors, which is shown in Equation 4.

12. Line 199: Why are the fourth and fifth right-hand terms are the same in Equation 5?

AR: Modified in Line 239.

13. Line 200-201: What do the annual mean precipitation (i.e., Xprecip) and temperature (i.e., Xtemp) refer to? Do they refer to the commonly used mean values of precipitation and temperature or the mean values of the precipitation and temperature aggregated to the aggregation level of the drought index (as mentioned in Line 172)? Clarify the "annual mean" here. Does Dτ also refer to the annual mean values of scPDSI or SPEI?

AR: For the linear model for scPDSI, $X_{precip}$ and $X_{temp}$ mean the commonly used annual mean, because the scPDSI cannot calculate the droughts for different time scales. For the linear models for SPEIs at different time scales, $X_{precip}$ and $X_{temp}$ should firstly aggregate analogously to the aggregation level of the SPEIs, then calculate the annual mean of precipitation and temperature. And $D_\tau$ is the annual mean value of drought indices. More explanations have been added in Line 241-243.

14. Line 228: What does $\hat{y}_i$ denote?

AR: It should be $\hat{Y}_i$ ,changed in Line 271.

15. Line 275: "ominates" might be "dominates"?

AR: modified in Line 320.

16. Line 320-321: High/low temperatures lead to a notable negative/positive response of SPEI03 to forest cover, instead of decrease/increase in forest cover.

AR: Changed in Line 371-372.

References:

Leite-Filho, A.T., et al. 2021: Deforestation reduces rainfall and agricultural revenues in the Brazilian Amazon. Nature Communications, 12, 2591.

Liang, Y., et al. 2022: Deforestation drives desiccation in global monsoon region. Earth's Future, 10, e2022EF002863.

Luo, X., et al. 2022: The biophysical impacts of deforestation on precipitation: results from the CMIP6 model intercomparison. Journal of Climate, 35(11), 3293-3311.

Smith, C., et al. 2023: Tropical deforestation causes large reductions in observed precipitation. Nature, 615, 270–275.

McCullagh, P. and Nelder, J. 1989: Generalized Linear Models, CRC Press, Boca Raton, Fla, 2 edn.

---

## Author Response (AR2)

Authors' Response to Reviews of

**Using statistical models to depict the response of multi-time scales drought to forest cover change across climate zones**

*Yan Li, Bo Huang and Henning W. Rust*

*Hydrology and Earth System Sciences (HESS)*

RC: Reviewers' Comment    EC: Editor' Comment    AR: Authors' Response

Dear Prof. Dr. Genevieve Ali,

We hereby submit the second revised version of our paper entitled "Using statistical models to depict the response of multi-time scales drought to forest cover change across climate zones".

We would like to sincerely thank the editorial office and the reviewers for the efficient handling of our manuscript and the constructive suggestions that helped to improve the quality of the paper. We have carefully addressed all the comments and a produced a revised version of the manuscript where they are considered. Changes following the comments of reviewers are marked in an independent file with track changes.

To address the reviewers' comments, we provide a more detailed explanation regarding grid-point-wise training and detrending analysis methods, and highlight the distinctions between these two methods in the revision manuscript. To clarify this aspect of our study, we also have adjusted the text accordingly.

Please do not hesitate to contact me should you need any more information.

Yours sincerely,

Yan Li

(On behalf of the co-authors)

**Editor**

EC: Two reviewers have assessed your revised manuscript. They appreciated the changes that you made, which add up to a more robust manuscript, but still raised a few comments for you to address, especially when it comes to **methodological approaches/statistical analyses**. Because at least one of the suggestions made by reviewers would require **a data re-analysis for comparison purposes**, I am returning your manuscript for moderate revision. Please note that upon reception, your revised manuscript will be sent back for review.

AC: We would like to sincerely thank your efficient handling and suggestion to improve this manuscript. Regarding the comment on grid-point-wise training and detrending analysis, we provide further answers and explanations on statistical methods. We use a generalized linear model (Eq. (5)), not a traditional linear model, which has an analysis of interaction (more explains in Line 235-238) and therefore cannot interchange averaging and model building. The grid-point-wise training work can be served as an extension of the present work (discussed in the text L501- 505). For detrending analysis, we focus on a statistical description of drought indices as functions of temperature, precipitation and forest cover, independent on external drivers of these variables. Figure B1 help us identify any underlying trends over time that may not be accounted by the existing predictor variables (temperature and precipitation) (more explains have been added in Line 288-292).

**Anonymous referee #1**

RC: The authors tried to answer all my concerns point by point. However, I was expecting more in detail analysis.

AD: Thanks for the useful feedbacks to improve the quality of the manuscript and we add more details in the analysis.

RC: show that the grid-point-wise training and the averaging over the categories lead to the same results (at least for one category);

AC: For the linear model in Eq. (5), it is not possible to interchange averaging and model building as there are interaction terms (more explanation about the interaction in L235-238). Hence the result from first building a grid-point-based model and then averaging the result will in general not yield the same result. The interpretation for these two approaches differ: The result from our approach quantifies the effect of aggregated forest cover change, temperature and precipitation on an aggregated drought index for a given climate zone.

Modelling individual grid-points is a lot more difficult as there are -- as you say -- very heterogeneous influences which we cannot grasp with the model presented. We focus on large scale effects here. (As an outlook, one could split up the climate zones into large regions and look for differences.)

And we also add some explanation "In our research, to simplify the initial study, we chose to aggregate data across different climate regions. This approach helps to smooth out localized variations and complexities. Going into a more detailed spatial analysis would be a deeper level of investigation. However, it's important to note that the conclusions might not be entirely consistent when transitioning to a grid-point-wise training approach. This inconsistency arises due to interaction terms in the model building process." (in L501-505)

RC: show that the detrending would lead to the same results as in the original time-series,

AC: For the detrending: the purpose of our model is to quantify changes in drought indices as results of changes in forest cover, temperature and precipitation. If we remove trends in the target variable (drought index) and the covariates (temperature, forest cover, precipitation), we would change the relation between the drought index and our covariates. For a more elaborated answer on detrending, see our last reply page 15.

**Anonymous referee #2**

RC: I would be happy to accept the manuscript. We thank the authors for their careful revisions to address my comments. I am overall satisfied with the authors' revisions and responses. The manuscript is much clearer than the original one, particularly for the Method section. I acknowledge that I do have a bias against using linear regression model to address the issue with multiple physical processes being fully coupled. Nevertheless, I think that the linear model used here is perhaps a useful tool to examine the effects of forest cover changes, as well as temperature and precipitation, on drought. I appreciate the authors' effort to add some explanations on their findings in the revised manuscript, but some results are still not sufficiently explained. Please see my detailed comments below.

AC: Thanks for the appreciation of our work and the useful feedbacks to improve the quality of the manuscript.

RC: Line 356-361: It sounds plausible that afforestation reduces PET via an increase in air humidity when precipitation is higher than above. However, I still cannot understand why afforestation increases PET when precipitation is lower than above. The authors attribute the increased PET to the water supply restriction. It is true for actual ET but not necessarily for PET. I am wondering whether the increased PET is linked to higher net radiation (according to the Penman–Monteith Equation), with the latter being further attributed to the lower albedo of higher forest cover.

AC: In this context, it's important to note that the temperature remains constant throughout the analysis (in the first and second rows of Figure 6). The change of net radiation is not involved. Consequently, the variations in drought conditions are solely influenced by changes in water supply. To clarify this aspect of our study, we have adjusted the text accordingly (Line 366-382).

We can see that the influence of forest cover on drought conditions varies depending on precipitation levels and geographical regions. For SPEI03, it appears that forest fraction has a relatively modest impact across various levels of precipitation (lines are close to be horizontal). This suggests that precipitation does not significantly modulate the influence of forest cover on the short-term drought index (the first row in Fig. 6). However, when we examine SPEI24, a different pattern emerges. There is a strong influence of precipitation on the forest fraction effect (the second row in Fig.6), particularly in arid regions (as seen in Fig. 6F). In general, as precipitation increases beyond the median level, the drought index tends to rise with increasing forest cover. This phenomenon can be explained by increased transpiration associated with larger amounts of precipitation, resulting in a reduced vapor pressure deficit (VPD) and, consequently, lower PET. This leads to higher SPEI24 values when forests are denser. Furthermore, forests have the capacity to intercept precipitation and diminish ground-level wind speeds. These combined effects contribute to a reduction in PET as forest cover increases. If precipitation is lower, this effect decreases and the slopes in Fig. 6F get smaller. There is not sufficient water to be evapotranspirated, even if the forest fraction increases. For a specific amount of precipitation (about the median) the slope is 0. When the precipitation is less than this amount, we see a negative slope suggesting the interpretation that for restricted water supply, an increase of trees leads to an increase of PET and hence to a decrease of SPEI24. An opposite effect can be observed in snow region (Fig. 6N). Here, with minimal forest cover, precipitation directly affects the SPEI24 leading to a more humid situation with higher precipitation. However, with forest cover increases, this direct effect vanishes. It's worth noting that for snow regions, the model captures less than 30% of the total variability, indicating the complexity of this relationship.

RC: Line 371-374: The authors find that higher temperature leads to negative responses of SPEI03 to forest cover increases. Their explanation is that "rising temperature leads to increasing transpiration from trees, and if there are more trees, more water will be taken away, and the drought indices will decrease". This explanation is not complete as the authors uses PET and P, instead of soil moisture, to measure drought. I think the full explanation should be more trees->higher evapotranspiration->less soil water->less evapotranspiration->higher VPD->higher PET->lower SPEI.

AC: Thank you for your suggestion. The precipitation remains constant throughout the analysis (in the bottom two rows of Figure 6). Yes, rising temperature leads to increasing transpiration from trees, and increasing transpiration will increase PET, then conducting a decline of drought indices. We revised the manuscript according your comment. "Elevated temperatures trigger greater transpiration rates from trees. When there is an abundance of trees, they collectively draw more water from the soil. This depletes the water content in the soil, and when soil moisture becomes insufficient, it results in reduced evapotranspiration. This decrease in evapotranspiration leads to a higher vapor pressure deficit, and subsequently, an increase in PET. This shift toward higher PET values often corresponds with a decrease in drought indices." (L390-394)